# China-Pakistan economic corridor and its impact on rural development and human life sustainability. Observations from rural women

Ahmad Saad[1]*, Mariah Ijaz[2], Muhammad Usman Asghar[3], Liu Yamin[1]

**1** Zhou Enlai School of Government, Nankai University, Tianjin, China, **2** Department of Sociology, Bahauddin Zakariya University, Multan, Pakistan, **3** Center for International Peace and Stability, National University of Sciences and Technology, Islamabad, Pakistan

\* ahmadsaad08@hotmail.com

## Abstract

China-Pakistan Economic Corridor (CPEC) has become a "game-changer" not only for policymakers but also for common citizens of China and Pakistan because of its potential benefits in the economic prosperity and sustainable development in the lives of individuals. Recently, scholars have shown a great interest in researching the eminence of CPEC. Despite prodigious efforts of scholars, the question "how CPEC would influence living standards of rural women and what they perceive?" has been remained unanswered. The present study is the first attempt to unearth the perceptions of rural areas' women towards CPEC. We collected empirical evidence through a structured questionnaire from 302 educated rural women and interviewing 32 uneducated rural women. The following major conclusions are drawn through structural equation modeling via AMOS 21: the rural women significantly directly perceive new opportunities through CPEC; however, they perceive that the CPEC would not directly influence their quality of life and self-enhancement, but through the development of these rural areas. Rural development partially mediates the relationship between CPEC development and perceived opportunities while it fully mediates the relationships between CPEC development and quality of life as well as between CPEC development and self-enhancement. Policymakers need to emphasize on development of rural areas that would improve living standards of poor communities. The government needs to meet the expectations of poor communities and rural women to ensure their sustainable development.

## 1. Introduction

China Pakistan Economic Corridor (CPEC) has become a central focus of research due to its significant role in sustainable development goals, economic development and living standards [1]. This is the largest economic corridor between China and Pakistan in the history of their

**Data Availability Statement:** We used a structured questionnaire and in-depth interviews to collect data from the rural women. The respondents were ensured that the data are only used for research

purpose. Therefore, restriction from the ethics committee of NUST is applied for the data access. The data (related to tables and figures) can be obtained on request from the ethics committee of NUST: anwar.msfin614@iiu.edu.pk.

**Funding:** This paper does not receive external funding.

**Competing interests:** The authors have declared that no competing interests exist.

mutual relationships. Both countries have their own goals to be gained through the CPEC, particularly Pakistan aims to overcome the challenges faced in economic development, energy and social welfare while China intends to extend its trade periphery in the global market and to create energy and trade routes in the future, aiming at high economic growth [2]. Most of the developing economies, especially Pakistan, face the major economic problems of unemployment, lack of resources and other social problems. Hence, it is perceived that the economic corridor would help to overcome these issues.

CPEC is considered as a "game-changer" for Pakistan and the Chinese economies in terms of trade, economic growth and development [3]. It would benefit both the rural as well as the urban population of both countries. This corridor encompasses different projects, and some of the major projects of CPEC have already been completed while other projects are still in progress (see Tables A, B, C, D and E for details in S1 Appendix). These projects are expected to overcome some of the social and economic problems faced by these underdeveloped regions of Pakistan. Considering the economic hardships, women in rural areas face several barriers such as wage discrimination, lack of business opportunities, poor health facilities, occupational segregation, and harassment in the workplace. According to a recent report of 2018, around 39% of women in Pakistan are subjected to spousal violence and face severe family and community sanctions. Therefore, rural women perceive a positive change in their lives through CPEC projects.

This research examines the perceptions of rural women in the context of social issues and focuses on how CPEC can possibly overcome these issues (such as, lack of opportunities, lack of self-enhancement and poor living standards etc.). The main objectives of the corridor are to; improve economic development, reduce poverty, promote free trade, encourage foreign investment and enhance living standard etc. [4, 5]. For instance, Haq and Farooq [6] reported that CPEC would improve social welfare by an average of 5.21% in Pakistan. They further identified that at the provincial level the social welfare in Baluchistan would improve by 6.4%, in Sindh by 6.31%, in KPK by 5.19%, and in Punjab by 3.5%. Additionally, they also provided the sector level estimates of these improvements and reported that it would improve the education sector by 3.85%, health sector by 4.74% and housing sector by 8.6% which would improve the living standards of the people. There are several goals connected with CPEC, however, sustainable development and social welfare were the most priority areas of these megaprojects [1].

To examine the benefits of CPEC, the existing research is diverse and revolves around two streams, micro-level targeting individuals' benefits [7–9] and macro-level targeting the country level benefits [10–13]. At the micro-level, the relevant studies focused on the quality of life, job opportunities, poverty alleviation and living standards in general [8, 14, 15]. However, research on the importance of CPEC in rural areas of Pakistan is very rare. In particular, how these initiatives would affect the life of common man and woman, and whether the effect was direct or indirect through some intermediaries still need to be examined. This study attempts to fill this gap by examining the direct role of CPEC in improving the lives of common people, particularly the women, as well as the indirect effects through rural development in the form of development of local infrastructure.

The novelty, as well as contributions of this study, are threefold. First, considering the existing living standards and facilities available to rural women in Pakistan, which is attributed to the less or no access to the basic facilities like education, hospitals, parks, industry, which contributes to the low level of income [16], this study contributes in examining the potential benefits for these women in the rural areas. Zakar, Zakar [17] reported that the non-availability of these opportunities further deteriorates their social status in the society and resultantly were less empowered [18]. The initiatives of the CPEC are expected to be useful in overcoming this social problem by enhancing their living standards through the provision of economic

opportunities [14, 19]. Second, our research provides some empirical evidence and contributes to the literature of CPEC, particularly in terms of rural development, infrastructure and the sustainable life of rural women. Previous studies discussed the aforementioned factors mostly in a qualitative way [19–21]. This study uses empirical evidence gathered from rural women living in the rural areas of Pakistan. Third, from a theoretical perspective, studies on CPEC are lacking and none of the studies has tested a theory deductively, particularly, this research contributes to the Theory of Community Development—initially proposed by Sanders [22] and expanded by Wilkinson [23] and the Social Capital Theory [24, 25].

We draw on the theory of community development (also referred as "field theory of community development), which sheds light on the actions (by internal and external bodies) purposively directed towards positively altering community structure [23]. Additionally, this research also contributes to the social capital theory [25] which argues that social integration facilitates the accumulation and development of human capital. This theory further states that rural communities build positive ties with external bodies, such as government, industries and NPOs to acquire resources for their betterment [26]. These theories are widely tested in western and developed economies' context while their effectiveness in the context of the emerging and under-developed economies were rarely debated [27, 28] and in particular, our study is the first attempt to unleash the importance of governmental projects in community development, in the context of the social capital theory and the theory of community development.

Emphasizing the policy implications, this study suggests implications for practicing managers, human rights departments, policymakers and the practitioners. CPEC is in progress and several policies are designed and implemented for the betterment of the rural communities. However, very limited policies are focusing exclusively the rural women such as facilities and incentives for starting a business, accessing health and education facilities. Therefore, this study draws the attention of policymakers towards improving the lives of rural women through CPEC. Moreover, this research also gives some recommendations to the government to work on the infrastructure of rural areas through the initiation of developmental projects of basic facilities targeting exclusively the poor and rural communities. Additionally, the present recommendations and implications can be applied in China to build a better infrastructure in their rural areas connected through the corridor. Furthermore, the implications of this research can be applied in the relatively less developed European economies where new policies and strategies may be introduced for rural community development. Finally, this research opens a new zone for academia and researchers who are engaged in researching rural development and sustainable goals.

This article is organized as follows: In the first section, we discussed the importance, objectives and contributions of this study. We reviewed the relevant literature which culminated problems of rural women and connections with CPEC and also developed relevant hypotheses in section 2. In section 3, we discussed the methodology explaining the sample and the collection of the data. In the fourth section, we described the analytic procedure used in this study and finally, we discussed the contributions and recommendations along with the implications and future research directions in the last section.

## 2. Theoretical underpinning

The objectives of this study were to examine the roles and the benefits of the CPEC initiatives in the rural areas of Pakistan, particularly for women. Particularly, the benefits associated with the women's living standards and the basic facilities available to them in terms of growth opportunities, quality of life and self-enhancement opportunities. We also assumed that these opportunities would come through the development that took place in the regions, hence a

mediating role of the rural development is also examined in transferring the effects of these projects to the common people. In this context, we drew on the theory of community development and the theory of social capital.

## 2.1 Theory of community development

The theory of community development, alternatively termed as the field theory of community [22] argued that there were different factors at the local, regional and national level which have some implications for rural community development. In addition, Wilkinson [23] also argued that developmental strategies were needed for building a sustainable community. This theory sheds light on the actions (by internal and external bodies) which were purposively directed towards positively altering community structure [23]. Sanders [22] further highlighted the importance of the practitioners and the social scientists in the community development. The formers are involved in the administrative side and more action-oriented, however, the latter are contributing theoretically and identified four possible ways in which these two actors can progress such as the process, method, program and movement (See Fig 1). *Process* implies that there is a need for interactions, actions and steps towards rural community development [23]. *The method* highlights the effectiveness in understanding the community through research, mobility, planning and organizing the resources for rural community development. *Program*s are the schemes, policies, and projects, which are initiated in order to develop the rural community. Finally, *Movement* emphasized the personal commitment between communities, the *practitioners* and *the social scientists*. *Practitioners* and *social scientists* should emotionally

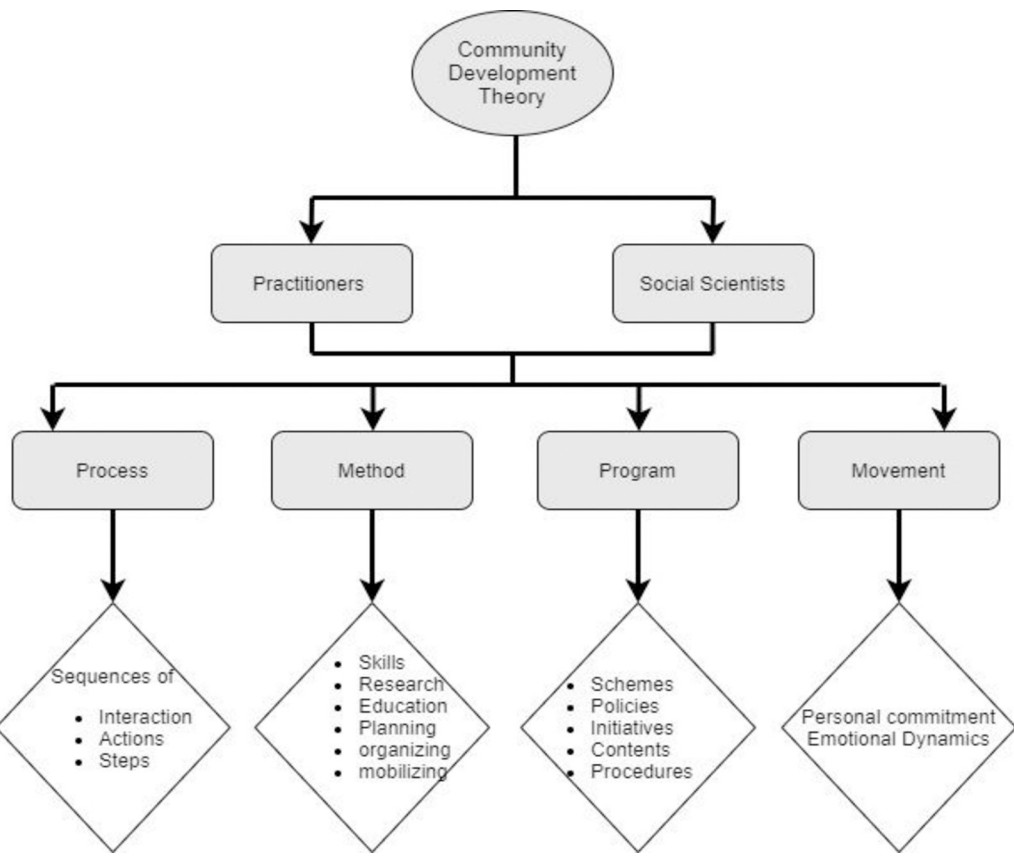

**Fig 1. Theory of community development (self-developed).**

attach to the community to achieve the desired targets of the programs initiated for the development of the communities. Globally, community development is achieved through acknowledgment of the present needs and the prevailing environment of that society. For instance, Etuk and Acock [29] argued that rural community development was achieved through prioritizing the interplay between social, economic, environmental factors to get positive results of these initiatives.

Douglas [30] demonstrated that local government was the key to rural community development (building physical infrastructure and healthy programs) in Canada. Herbert-Cheshire [31] stated that in Australia the same strategies and governmental styles were adopted for the rural community development as British and Europe where individuals were promoted through developmental work to initiate their own projects of reform. Cramb and Wills [32] preferred small programs, policies and schemes (agriculture and local business promotion) for developing rural communities in Malaysia over large programs (large industries) and initiatives. Wahid, Ahmad [33] suggested that local government incentives and supports were essential for removing the barriers in the way to rural community development (specifically the quality of life and wellbeing) in Pakistan. Songling, Ishtiaq [34] suggested that the government's financial and non-financial incentives were important for industrial growth and business expansion in Pakistan. Drawing on these empirical evidences we argue that both local and federal level projects are important for rural community development in Pakistan.

## 2.2 Social capital theory

The underpinning principle behind the social capital theory is the social relationship, which is considered as one of the resources that facilitates the accumulation and development of human capital. This theory further argues that rural communities build positive ties with external bodies (government, industries and NPOs) to acquire resources for their betterment [25] which emphasize on the importance of relationships with the government, business industries and political bodies in terms of trusts, volunteerism, social mobility and togetherness etc. as shown in Fig 2. Several studies have shed light on the importance of rural community relationships with external bodies such as government, NPOs and industries for resource acquisition and rural development [28, 35]. Wakefield and Poland [36] argued that social capital cannot be conceived in isolation from economic and political structures since social connections were contingent on, and structured by, access to material resources. They further argued that the social ties of the community were based on the common interests of the community development and capacity building in rural areas. Alfred [26] appraised the social capital of rural women (e.g. with external bodies such as government and political bodies) for acquiring necessary material and access to the requisite resources for quality life and wellbeing. Wu and Liu [37] highlighted the importance of social capital in rural community development in the context of government projects and highlighted the important and key role the social capital plays in the transformation of government attention towards rural community development. Social networks have the potential to be leveraged by rural communities in pursuit of sustainable development efforts. Similarly, rural communities in Pakistan perceive several benefits when they build ties and relationships with the government and industries [27, 38].

The research model developed for this study to examine the hypothesized relationships between the variables is provided in Fig 3. CPEC development is the independent variable, in the middle, rural development mediates the relationships between the CPEC development (the independent variable) and the three dependent variables, such as opportunities, quality of life and self-enhancement.

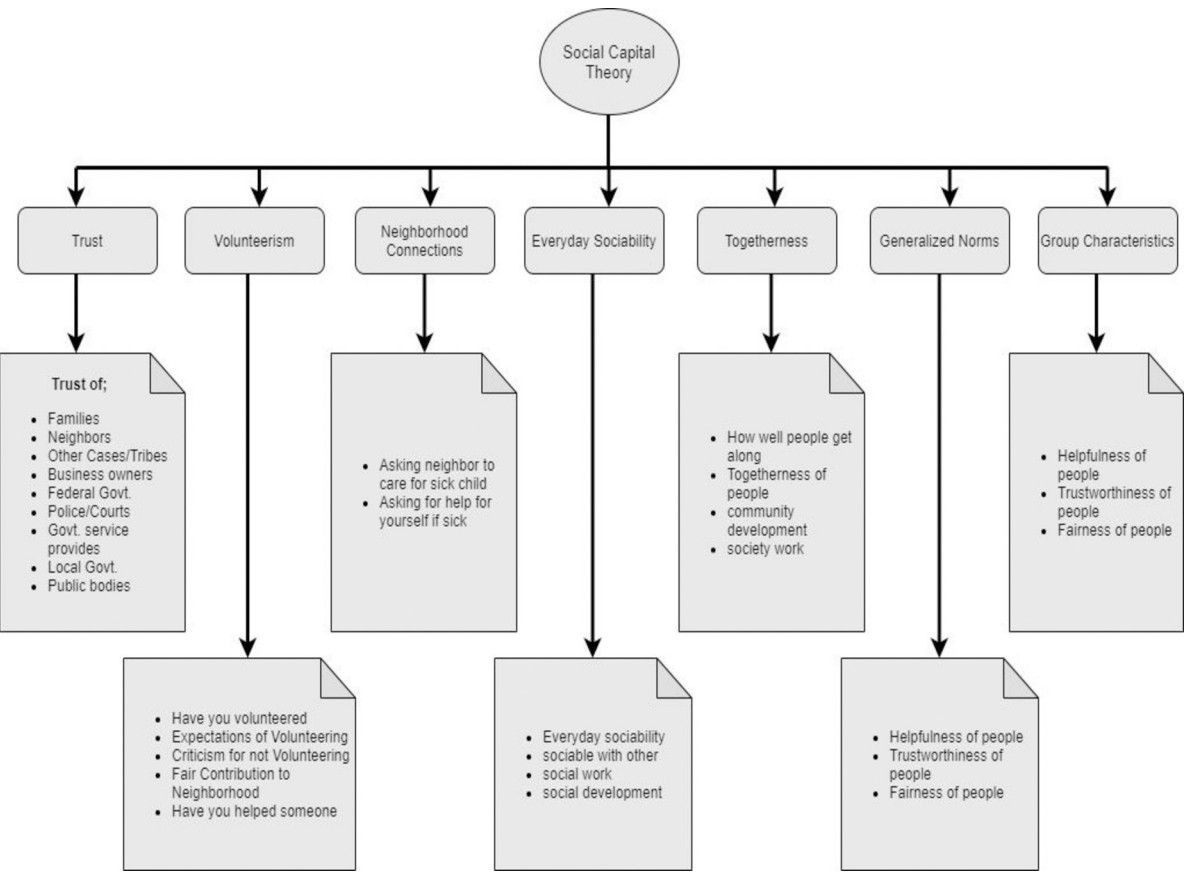

**Fig 2. Theory of social capital (self-developed).**

## 3. Rural women and CPEC projects

In Pakistan, women, no matter what role they play, either directly or indirectly contribute towards the socio-economic activities and economic growth of the country. Indirectly, they take care of family throughout their physical and mental development as housewives. Hence, empowering women in terms of education, health and other social awareness is necessary for

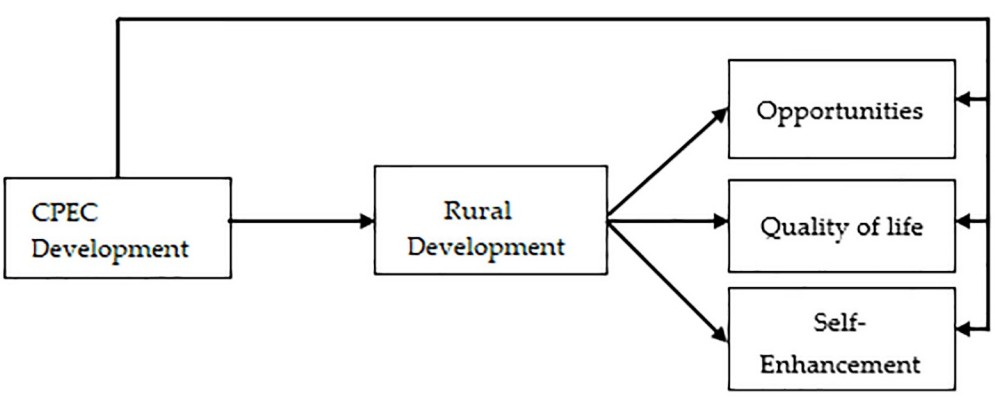

**Fig 3. Research framework.**

children's strong upbringing, physical development and sound mental health. Unfortunately, in the context of Pakistan, which is considered as patriarchal and termed as a male dominant society particularly in the rural areas, women are largely restricted to the households. Women are largely neglected and are considered as middle and upper sections of the society, and are subject to violence in the rural areas. Surprisingly, around 75% of women population of Pakistan live in tribal and rural areas [39]) and remained structurally disadvantaged and considered a second class citizen in the context of rights and decision making. Women are victimized both socially and culturally due to discrimination in the rural areas due to the traditional norms, family rules and low status in the society [39]. Women are subject to have experienced violent behavior in terms of rape, burning, honor killing and acid throwing, in particular, women in rural areas of Pakistan face economic and social discrimination [40]). Every year, several cases of rape, honor killing and other domestic violence are reported in Pakistan, which was a clear indication of the failure of enforcement of the relevant laws. For example, the constitution of Pakistan and other different laws prohibit these activities against women but authorities often do not enforce properly. For example, the property laws in Pakistan empower the women to have their rights in the properties, however, some social pressures from the family members and the traditions discourage them to raise their voices for their rights ([41]). Women empowerment is the only solutions for these problems, however, to empower them it is important to educate them, which is only possible through the provision of education and other subsequent benefits which are related to their employment and autonomy [42, 43]) highlighted the problems which the rural women face, for example, firstly, ignorance regarding their rights and their basic rights which can give them ease from doing the hard tasks of fetching water from far areas, fetching woods for burning, secondly, looking after the livestock of the household, and finally, they are often deprived of their social and economic rights such as health and education facilities, family planning measures, decision making authority and ownership of land and livestock along with the rights of their sale and purchase.

According to the new Census of Pakistan (2018), a total of 63.33% of the population (64% women) live in rural areas in the four provinces namely; Sindh, Baluchistan, Punjab and Khyber Pakhtunkhwa. Rural areas are those areas where the basic facilities such as hospitals, schools, industries and irrigation etc. are lacking, with relatively poor housing facilities. About 35% of Pakistani rural population live below the subsistence level where social services and basic facilities are absent. Women are more disadvantageous as compared to men in these rural areas, due to the cultural and traditional conventions. The most common problems faced by rural women are; low or no employment opportunities, high wage discrimination, lower access to socio-economic opportunities, lower access to health facilities, occupational segregation, and harassment in the workplace. Therefore, most of the people living in rural areas tend to migrate to urban cities to access the basic facilities and resulted in the population shift from rural to urban areas. For instance, Fig 4 shows the reduction in a rural population from 65.7% in 2008 to 63.3% in 2018.

To overcome the major issues in rural areas, the government has planned to encapsulate several programs in CPEC mega projects.

Several projects are initiated under the CPEC in the four provinces; Punjab, Sindh, Balochistan and KPK to benefit the urban and rural areas. We have discussed a few major projects that were intended to benefit rural areas of the four provinces.

## 3.1 Sahiwal 2x330MW coal-fired, power plant, Punjab

It is an energy project with 1,912.2$ million investment which would generate electricity with the coal energy and also includes building a railway track connecting the village of *Yusuf Wala* to the project site for the exclusive use of the power plant. This project would be beneficial for

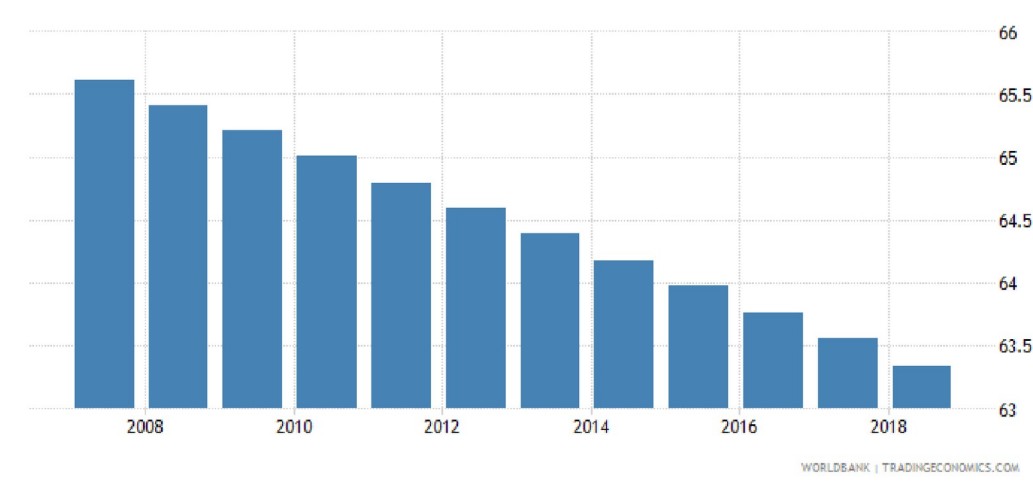

**Fig 4. Trend of rural population in Pakistan.**

the people of *Sahiwal* and *Okara* districts of Punjab province. In other words, the project is connected with CPEC and mainly aimed to overcome the electricity shortages in the areas, resulting in the creation of business opportunities for micro-businesses.

### 3.2 Rahimyar Khan coal power project, Punjab

This project is estimated to invest 1600$ million and expected to supply water and power (coal imported) to Rahimyar Khan, Punjab. Most of the people are engaged in farming and bricks Kiln in the area which requires more water. On the other hand, water shortage is the major issue of the people in this area which would be overcome through this project.

### 3.3 Quaid-e-Azam 1000 MW solar park, Bahawalpur, Punjab

It is a solar power project with two phases; in phase 1, Installed Capacity of (MW) 400 with 520$ million investment and phase 2, Installed Capacity of (MW) 600 with 781$ million investment. It would benefit the people of Bahawalpur in terms of planting solar panels in a wide range as an alternative source of electricity to overcome the electricity shortage in the district. Pakistani folks are more entrepreneurial, however, load shedding was the major hurdle. Hence, this CPEC project is expected to provide electricity to households as well as for businesses in the area, contributing to the regional as well as national economic growth. Fig 5 shows the progress overview of this project.

### 3.4 Karot Hydropower station, AJK & Punjab

This project is situated in the dual boundary of District Rawalpindi, Punjab & District *Kotli*, AJK, River *Jhelum* with an estimated investment of 1698.26$ million. This project includes the construction of concrete gravity 95.5 meters high dam with a crest length of 320 meters near the village of *Gohra*. The project would result in the construction of 72 homes and 58 local businesses in the areas.

### 3.5 Muzaffargarh coal power project, Punjab

This project has an estimated investment of 1600$ million with a major goal of improving water and power systems in the regions to benefit the people. *Chenab* River is located near to the city

| Project | Project Name | Status | Completion |
|---|---|---|---|
| 1 | 2x660MW Port Qasim Coal-fired Power Plant | Under Construction | 100% |
| 2 | 2x660MW Sahiwal Coal-Fired Power Plant | Two units inaugurated | 100% |
| 3 | Thar Coal-fired Power Plant and Surfice Mine in Block II of Thar Coal Field | Under Construction | 60% |
| 4 | 50MW Dawood Wind Farm | Commercial Operation Date achieved | 100% |
| 5 | 900MW Quaid-e-Azam Solar Park in Bahawalpur | Energization achieved | 100% |
| 6 | 100MW Jhimpir Wind Farm | Under Construction | 100% |
| 7 | 873MW Suki Kinari Hydropower Project | Under Construction | 65% |
| 8 | 50MW Sachal Wind Farm | Commercial Operation Date Achieved | 100% |
| 9 | 2x660MW Rahimyar Khan Coal Power Plant | Feasibility Stage | 15% |
| 10 | Thar Coal Block I and 2x660MW Mine Mouth Power Plant | To be inaugurated | 50% |
| 11 | 2 x 660MW Hubco Coal Power Plant | Implementation Agreement Initiated | 50% |
| 12 | 300 MW Gwadar Power Plant | Feasibility Stage | 60% |
| 13 | Matiari-Lahore Transmission Line | Negotiation in Process | 15% |
| 14 | Matiari-Faisalabad Transmission Line | Negotiation in Progress | 15% |
| 15 | 2x660MW Gaddani Powerplant at District Lasbela, Balochistan | Feasibility Stage | 15% |
| 16 | 2x660MW Muzaffargarh Coal-fire Power Plant | Feasibility Stage | 15% |

**Fig 5. Descriptions of the projects.**

but the government had not initiated any beneficial work on the bank of the river. This is the first time to provide electricity to the people of *Muzaffargarh* through Coal Power Project of CPEC. The following are the activities planned under this project. Construction of sub-stations:

- Installation of generators
- Construction of powerhouses
- Construction of related infrastructure

### 3.6 Port Qasim electric company coal-fired, 2x660, Sindh

This project was developed to overcome the electricity shortage in *Karachi*, *Sindh*. This project is also expected to provide electricity to the other cities and rural areas of *Sindh* to enhance

their livelihood and wellbeing. There are several opportunities in the bank of the river of *Karachi*. However, load shedding mitigates and hinders the progress of business operations and new initiatives.

### 3.7 Surface mine in Block ll of Thar coalfield, 3.8 metric ton per annum (mtpa), Thar, Sindh

This project (estimated cost of the project is 1,470$ million) was initiated especially for the Thar people—one of the badly affected areas of Sindh in terms of food and water shortages. The main goal of this project is to provide cheap electricity to the people of *Thar* for daily activities and cost-effective business initiatives. Another phase of this project focuses on environmental and climate safety measures to overcome the environmental hazards of developmental projects.

### 3.8 SSRL 2x660MW mine mouth power plant, Sindh

It is a power plant (Block I) with an estimated investment of 1912.12$ million. This project is also initiated to benefit *Thar* community which are suffering from natural disaster, food and water shortage. Two major goals "water and electricity" to be gained through this project—thereby providing many opportunities for the rural community.

### 3.9 Thar mine-mouth Oracle, Thar Sindh

This is the third project initiated to facilitate the Thar district of Sindh (Block VI). Though the estimated budget of this project is not yet finalized it is planned that this project would improve the infrastructure such as roads, hospitals and schools of Thar to create opportunities for the people.

### 3.10 Dawood 50 MW wind farm, Bhambore, Sindh

This project is being developed near to the city of *Banbhore*, *Sindh* with an estimated budget of 112.65$ million. It is expected that the energy provided through this project would be sufficient to provide electricity to 100,000 households in the local area.

### 3.11 UEP 100MW wind farm, Jhimpir, Sindh

This project has an estimated investment of 250$ million and was one of the top 14[th] priority projects of the CPEC. This project is intended to promote businesses connected with oil and gas, particularly in the areas of *Jhimpir*, *District Thatta*. This project has already started benefitting the local community by employing 850 local Pakistani in different sub-projects. Most of the employments are provided to poor people as a labor force. After realizing the problem of drinking water, the project team has decided to construct a drinking water plant for the local village. Moreover, the project team has also created a book bank for the local schools to promote reading culture in the area.

### 3.12 HUBCO coal power plant 1x660 MW, Hub Baluchistan

This project has an estimated investment of 1912.2$ million in the energy sector to provide electrical power to the area of *Hub*, *Lasbela* district of *Baluchistan*. The main goal of this project is to provide electricity on a long term basis to the area of *Hub*, *Baluchistan*. The project has targeted *Hub* river of *Baluchistan* to generate electricity for the local areas. Additionally, the project team is also engaged in several social developmental activities such as hospitals, schools, health and safety management and environmental safety in the local area.

### 3.13 Suki Kinari Hydro Power Station, KPK

This project is initiated in the River *Kunhar* (a tributary of River *Jhelum*) with a budgetary plan of 1707$ million. The main theme of this project is to supply electricity to the connected areas to boost business activities. *KPK* is the province where more than 60% of operated businesses were closed down [44]. Hence, supplying adequate electricity is need of the day for current and potential business.

The progress and the existing status of some of the CPEC projects are shown in Fig 5.

In addition to these projects, CPEC roads are crossing several rural cities such as *Issa Khel*, *Lakki Marwat* and *D.I. Khan* overcoming inter-district transportation problems. It is clear from the detail of the different projects that CPEC would benefit the rural areas both by providing direct benefits in the form of employment opportunities, and access to other socio-economic facilities, but also indirect benefit by generating entrepreneurial opportunities for the local businessmen.

## 4. Literature review and hypotheses

### 4.1 CPEC and sustainable life of rural women

CPEC is considered as one of the most beneficial game-changers for the economic development of the participating countries, such as Pakistan and China. The former is benefitting from the domestic infrastructure development, which would potentially help the local communities to improve their quality of life. Particularly, the initiatives are equally beneficial for the women in the rural as well as the urban areas of Pakistan in the form of enhanced economic opportunities for the women [45]. The benefits of CPEC are not only restricted to a specific region or area but the benefits can be experienced throughout the length and breadth of Pakistan [46]. In the context of Pakistan, the rural population, particularly the women are neglected in terms of job opportunities, education and health facilities, resulting in hardships for them, such as high transportation costs in accessing these facilities in far areas (Hussain, Zhu [47]. The local government agencies initiate different developmental projects for rural development, but very often fail to complete due to some political as well as economic reasons (Rafique and Rehman [48]. Mohmand, Wang [35] highlighted the importance of investment in technological progress, industry and infrastructure instead of targeting only a particular factor to comprehensively develop these neglected areas. However, the majority of the projects in Pakistan do not consider livelihood and social norms of poor people and rural areas [49]. However, the projects planned under the umbrella of CPEC comprehensively consider the infrastructure to affect different aspects of the lives of common people living in the rural areas through the use of transition technology to articulate things in a better way.

CPEC is expected to enhance the social goals of the people in terms of quality life, living standard and job opportunities. Hence, it gives recommendations to the policymakers for considering the environmental aspects while devising the strategies and programs in different regions [15]. It is also suggested for China and Pakistan to create safe and effective investment policies to encourage stakeholders and let them invest in different industrial projects to provide the basic facilities such as health, education and living standards [50]. Alternatively, responsible authorities and policymakers need to reduce the risk for investors in different regions and should encourage them towards high investment [51].

For example, the CPEC's Coal Power Plant Project Initiative (that are not particularly related to sustainable goals) but are expected not only to improve the lives of the women associated with farming but also to create more opportunities for them in different plants and projects [14]. Other similar megaprojects also have the potential to benefit society by improving

the socio-economic conditions of men and women in different regions [11]. CPEC is perceived to bring long-lasting advantages for both the Chinese and Pakistan economy which would ultimately be reflected in the improvement of the lives of the people by having more opportunities for employment and social integration through the mega projects [52]. Through CPEC, local people perceive unlimited business opportunities and intend to expand their business across the globe by receiving sufficient foreign credit and financial resources [53]. An empirical study of Chinese and Pakistan individuals reports that the projects of CPEC would potentially enhance the social benefits and would improve the living standards of people living in China and Pakistan [4].

In the context of the community development theory and the social capital theory, the extent to which the parties achieve their targets is subject to the mutual agreement between the parties [5]. Such as in the context of the CPEC, the expectation of the local people is an improvement in their quality of life. Similarly, the local communities are also expected to facilitate the policymakers and the project officers in their policies and work, to capitalize on the opportunities created by these mini-projects. In the absence of cooperation between the stakeholders, which would result in a lack of interest and can negatively influence the local community, for example, deterioration in the quality of water and other environmental pollution caused by the industrial projects [54, 55]. CPEC is favorable for neighboring regions but in particular, Pakistani citizens can get maximum benefits of it by initiating new businesses and promoting entrepreneurial activities, working on infrastructure, health sector and energy development [56]. The local scenic areas of Pakistan such as *Hunza* and *Gilgit Baltistan* would get the attention of the government by promoting tourism through the development of tourist attractions of tourists through with the help of these megaprojects of CPEC [57]. CPEC projects would help the local community to access the educational facilities, starting new businesses and other opportunities for social integration [58]. Local citizens in different areas of Pakistan perceive new profitable businesses with independent business profit to serve their families. These human development are the targets postulated in the sustainable development goals set by Pakistan, which would be achievable with the help of these projects planned under CPEC [59]. Considering the benefits of CPEC in sustainable development goals in Pakistan, it is argued that emerging economies need a more meaningful paradigm. However, CPEC can be the best source of attaining these goals if the policies are configured effectively for social, economic and environmental goals [1]. In terms of social benefits for community people, Naz, Ali [10] argued that CPEC would facilitate small businesses due to radical reduction in the logistic costs and time through the provision of better infrastructure for efficient transportation of goods. Moreover, it would create new employment opportunities for local communities, enhance industrial competitiveness for the local businesses and ultimately boost economic growth.

Hence some of the megaprojects of CPEC have already been started in different areas of Pakistan, therefore, the connected areas and regions have already started to see positive changes in their livelihood and life [60]. Local communities in China and Pakistan would easily get access to big cities in the respective countries and would be able to connect easily with urban areas. Therefore, they would access useful resources and information that are necessary for business growth [61]. In other words, people living in local and small villages would have more opportunities to interact with big and advanced cities through CPEC mega projects. Hence, these initiatives would enable the local communities to start their businesses thereby increasing more economic activities which would enable them to access basic facilities such as health and education [62]. Therefore, we hypothesize the relationships as follows;

*H1. CPEC has a significant influence on perceived opportunities creation among rural women.*

*H2. CPEC has a significant influence on the perceived quality of life among rural women.*

*H3. CPEC has a significant influence on perceived self-enhancement among rural women.*

## 4.2 CPEC, rural development and sustainable life of rural women

The perceived economic corridor envisioned under the CPEC was both advantageous as well as disadvantageous for the local community and people in urban and rural areas. In Pakistan, rural areas are the most neglected areas in terms of the development, which are due to urban-centric policies of the elected governments. However, due to the strategic importance of these areas, the major routes of the CPEC are passing through the rural areas of Pakistan. Hence, it would enhance the infrastructure of these rural regions, thereby resulting in a positive change in people's lives [63]. Despite bestowed with the natural resources, Pakistan's power and energy sectors are not performing optimally. CPEC is perceived to configure and boost the relatively underperforming sectors of the economy which can help Pakistan gain a strong power position in different regions [64]. Through CPEC projects, the local communities see various benefits including infrastructure development, transport facilities and tourism development in the areas which in turn would enhance their living standard [65]. However, the disadvantages can be few employment opportunities for local people in the industries because of the lack of the skills set required to undertake these jobs. Hence, they would not be able to secure the desired employment in the relevant industries [51]. Similarly, Saad, Xinping [15] also stated that CPEC can potentially harm the environment because of the mega infrastructural projects, however, they can be overcome by providing necessary safety training and employing the environment-friendly practices during work. Policymakers have committed to ensuring that CPEC would radically improve the lives of the poor communities and help in poverty alleviation enabling them to enjoy a good and satisfactory life [58]. This would create several opportunities for them such as jobs, business, health and education which would result in better outputs in their living standard [60].

CPEC is expected to encourage foreign investors and foreign direct investment in different industrial sectors of Pakistan. In this manner, the local communities in Pakistan would enjoy employment opportunities and business growth opportunities [12] and would result in not only in the development of the local infrastructure, but also the economic development of the country as a whole [66]. Naz, Ali [10] claimed that CPEC would boost industrial growth and would link business industries of Pakistan globally which would benefit households' social welfare. These international linkages would result in more FDIs and high economic growth for both China and Pakistan [67]. The megaprojects would positively contribute to social welfare and sustainability by improving the infrastructure [68]. For instance, CPEC would configure the developmental projects (infrastructure, the tourist industry, housing, hospitals, schools, food, livestock, energy and social welfare) in *Kashmir* that is a moderately developed area of Pakistan and a central destination of tourists [69].

With the announcement of CPEC projects, local people and communities have observed and perceived a great improvement in health facilities, job opportunities and infrastructure [9]. In particular, CPEC routes connect rural and urban areas of Pakistan which would increase the access of the people from rural areas to the urban areas to access the basic health, education and social-economic facilities [70]. Similarly, tourists have perceived a very positive change in the infrastructure of tourists' places in different areas of Pakistan because of the mega projects of the CPEC. Hence, by increasing the number of tourists, business growth and performance of the local communities would improve [19]. CPEC will promote not only the mutual trade relationship between China and Pakistan but also with other countries across the

globe. Hence, the neglected areas should be developed to become a central focus for business which in turn would benefit local communities and their living standards. CPEC is expected to reduce the geographical distance between China and Pakistan as well as with other neighboring countries that would result in extended trade and economic development [13]. Drawing on these assertions, we develop following hypotheses for this study;

*H4. CPEC has a significant influence on rural development.*

*H5. Rural development has a significant influence on perceived opportunities creation among rural women.*

*H6. Rural development has a significant influence on the perceived quality of life among rural women.*

*H7. Rural development has a significant influence on perceived self-enhancement among rural women.*

*H8. The rural development mediates the relationships between the development of CPEC and perceived self-enhancement among rural women in Pakistan.*

*H9. The rural development mediates the relationship between the development of CPEC and perceived quality of life among rural women in Pakistan.*

*H10. The rural development mediates the relationship between the development of CPEC and perceived self-enhancement among rural women in Pakistan.*

## 5. Methodology

### 5.1 Data and sample

This research used a mixed-method approach (structured questionnaire and interviews) in a sequential way (first we collected data through a structured questionnaire and then conducted some in-depth interviews). The participants in this study are the women in the rural areas of Pakistan. Women from rural areas are an important population, as in the context of CPEC, the benefits of the CPEC are under-researched, particularly from the perspective of the women. We prepared a questionnaire and an in-depth interview for data collection that were approved by the research committee of the National University of Science and Technology (NUST) for data collection. To get the approval of the ethical committee, we presented them with the written questionnaire with necessary information (cover letter, demographic information of individuals and questions of the main variables). After minor comments and discussion, they approved the survey for data collection. In other words, this research is approved by the research committee of the National University of Science and Technology (NUST), Pakistan. We have printed the questionnaire (attached in the appendix of the manuscript) that is approved by the research committee of NUST for data collection. Using the structured questionnaire, we collected data from the women above 20 years of age and have at least matric level qualifications, as they were expected to have some basic knowledge about the CPEC activities. For instance, educated people have more knowledge and understanding of CPEC and other mega projects and are able to predict future benefits. However, to articulate the results in a better way, we also gathered information from 32 illiterate women. We interviewed only those illiterate women in rural areas who were aware of the CPEC. Since it was difficult to know the exact number of educated women in the areas. CPEC is initiated recently, it is a new project and is not a general project like others. Therefore, is argue that in general, an illiterate person may not able to understand and predict the benefits of the CPEC. During the survey

and interview, we also acknowledge that most of the illiterate women do not know the CPEC. So it was difficult to receive relevant information for our research. Hence, we only interview those illiterate women who have some sort of knowledge of CPEC. Therefore, we used a convenient sampling approach to complete the data collection. The convenient sampling method was also appropriate due to the wide geographic distance in these areas. Hence, both educated (for questionnaire) and illiterate (for interview) women were approached conveniently. Moreover, the respondents were informed in starting of the survey that "the survey is a volunteer and your information can help us in completing the research". We used a structured questionnaire for data collection because it gives a maximum response rate in emerging economies, especially Pakistan. The language of instruction was English in the survey, as many Pakistani easily understand the language. However, where the respondents faced problems in understanding the concept, we facilitated them in translating the meaning and context. We selected only one participant from each household and ensured them about the data secrecy in the cover letter of the questionnaire. We distributed 700 questionnaires in the rural regions; *Lakki Marwat, D.I. Khan and Issa Khel*. The CPEC route passes through three regions. Within two months, we received 302 (43.14% response rate) accurate responses that were used to estimate the relationships between the variables.

Descriptive statistics of the sample are provided in Table 1. There, 43 women who have age range between 20 to 30 years, 77 women were between 31 to 40 years, 74 women were between 41 to 50 years old, 64 women were between 51 to 60 years old while 44 women were over 60 years. 63 women have completed matriculation, 98 women were secondary level while 71 were having a bachelor's degree, 55 were having a master degrees, 13 were MPhil degree holders, and only 2 were PhDs.

For the interview, we contacted more than 70 uneducated women, however, most of them were unaware of the CPEC, and only 32 had some knowledge of CPEC. We interviewed all the 32 women. A total of 19 women were between 20 to 30 years old, 10 women were 31 to 40 years old and 3 women were above 40 years.

### 5.2 Measurement of the constructs

*CPEC development*: Since CPEC is in progress, and it is hard to quantify its benefits. However, previous research considered perceived benefits as the reflections of the future benefits [5, 7,

**Table 1. Background detail of the rural women.**

| Particulars | Frequency | Percentage |
|---|---|---|
| Age of the Respondents | | |
| 1. 20–30 Years | 43 | 14.2 |
| 2. 31–40 | 77 | 25.5 |
| 3. 41–50 | 74 | 24.5 |
| 4. 51–60 | 64 | 21.2 |
| 5. 61 and above | 44 | 14.6 |
| Educational Background | | |
| 1. Matriculation | 63 | 20.86 |
| 2. Secondary School | 98 | 32.45 |
| 3. Bachelor | 71 | 23.51 |
| 4. Master | 55 | 18.21 |
| 5. MS / MPhil | 13 | 4.30 |
| 6. PhD | 2 | 0.70 |
| Total | 302 | 100 |

12, 13, 67], therefore, we also considered the perceived benefits of the CPEC, and adopted a 5-item tool to measure CPEC development (Saad, Xinping [15]. **Rural Development** was measured through a six-item tool adopted from Baig et al. (2018). The tools measure improvement in health facilities, education, electricity, agriculture, food etc. **Opportunities creation** was measured by 5 types of perceived opportunities such as "CPEC will provide employment and skills for improved livelihood opportunities" and "CPEC will generate new job opportunities in the rural sector". These items were adopted from Saad, Xinping [15]. **Quality of life** was measured by four items tool adopted from Saad et al. [15]. **Self-enhancement** was measured by using 5-item tool adopted from Thyroff and Kilbourne [71]. However, for significant matching, we slightly modified the items where respondents were asked: "to what extent, do you agree or disagree with the following statements through the development of CPEC will enhance?". For instance, a representative item is "4. Authority (the right to lead or command)". Five-point Likert scales were used in the structured questionnaire displayed strongly disagree = 1 to strongly agree = 5.

## 5.3 Common method variance

We used Harman's single factor test in SPSS using the principal component analysis to see the common method bias. The results provided 5 factors that have eigenvalues greater than 1 and the first factor explained only 27.083% variance that is below 50%, which ensured non-prevalence of the common method bias. We also check the influence of a common latent factor on the confirmatory factor model in AMOS. We matched the results of both measurement models and revealed that the model with a common latent factor is not good fitted. Hence, this test also confirmed the absence of common method variance in the data. Besides, we also tested the data for non-response bias. To see the non-response bias we compared the first and late responses [72], and the results show that there is no difference between the early and late responses for all the constructs of the study, hence the data is free from non-response bias.

## 5.4 Descriptive statistics and correlation

The descriptive statistics are provided in the Table 2. The results show that all the variables are normally distributed because the skewness & kurtosis were below the ±1 as suggested by Anwar, Rehman [73].

The Table 3 provides the correlation matrix. The results show that CPEC is significantly positively related with rural development (r = 0.411, p < 0.05), opportunities (r = 0.393, p < 0.05) and self-enhancement (r = 0.143, p < 0.05). In other words, we found a moderate correlation between CPEC and rural development, CPEC and opportunities and a weak correlation between CPEC and self-enhancement. However, relationship of CPEC with quality of life is insignificant (r = 0.098, p > 0.05). Rural development is significantly related with opportunities (r = 0.517, p < 0.05), with quality of life (r = 0.186, p < 0.05) and with self-enhancement (r = 0.371, p < 0.05). The results are matched with previous studies where moderate and

**Table 2. Descriptive statistics.**

| Variables | Mean | Std. Deviation | Skewness | Kurtosis |
|---|---|---|---|---|
| CPEC development | 2.7620 | 0.31900 | 0.342 | -0.099 |
| Rural Development | 3.5438 | 0.41813 | 0.284 | -0.092 |
| Opportunities | 2.2206 | 0.27314 | 0.287 | 0.149 |
| Quality of Life | 1.6079 | 0.36068 | -0.256 | -0.644 |
| Self-Enhance | 3.2461 | 0.52741 | -0.340 | 0.035 |

**Table 3. Correlations matrix.**

| Variables | Age | Education | CPEC | RuralDevelop | Opportunities | QualityLIfe | SelfEnhance |
|---|---|---|---|---|---|---|---|
| Age | 1 | | | | | | |
| Education | 0.010 | 1 | | | | | |
| CPEC | 0.166** | 0.268** | 1 | | | | |
| RuralDevelop | 0.049 | 0.199** | 0.411** | 1 | | | |
| Opportunities | 0.163** | 0.171** | 0.393** | 0.517** | 1 | | |
| QualityLIfe | -0.072 | 0.165** | 0.098 | 0.186** | 0.187** | 1 | |
| SelfEnhance | 0.003 | 0.192** | 0.143* | 0.371** | 0.307** | 0.179** | 1 |

** Correlation is significant at the 0.01 level (2-tailed).

*. Correlation is significant at the 0.05 level (2-tailed).

weak correlations are reported between the development of CPEC and perceived community benefits and opportunities [58]. Moreover, there is no multicollinearity issue in the sample data because all the correlations values are below the cutoff range of 0.80 [74].

## 6. Data analysis and results

We estimated the model using structural equation modeling through AMOS 21. We executed confirmatory factor analysis to ensure the fitness of the model, validity and reliability and structural model for testing the hypothesized relationship.

### 6.1 Confirmatory factor analysis

Standardized factor loading, validity and reliability of the items as well as constructs are provided in Fig 6. Fitness of the measurement model indicate a satisfactory fit (CMIN/DF = 2.047 which is less than 3) [75], furthermore, GFI = 0.88, AGFI = 0.85, TLI = 0.92, CFI = 0.93 and NFI = 0.88 are closed to 0.90 display a good model fit [76]. RMR = 0.017 and RMSEA = 0.059 also exhibited desirable fit [75]. We calculated the composite reliability of the constructs and found that all the variables have their composite reliability above 0.70. It shows satisfactory values as per the recommendation of [77]. The convergent validity of all the constructs were established as all the constructs' AVEs were greater than 0.50 [78]. For discriminant validity, we compared the square root of AVE which should be greater than the inter-correlation of the construct with other constructs [79]. Therefore, the discriminant validity of the constructs is established. The results are provided in Table 4.

### 6.2 Structural model

We estimated the structural model to test the hypotheses of the study (Fig 7). The model fit indices showed perfect fit of data to the model as all the cutoffs; CMIN/DF = 2.058, GFI = 0.88, AFGFI = 0.85, CFI = 0.93, TLI = 0.92, NFI = 0.0.88, RMR = 0.021 and RMSEA = 0.059 were in the acceptable range [75].

The results (Table 5) show that CPEC has a significant influence on opportunities ($\beta$ = 0.201, $p < 0.05$) which supported H1. Our findings favor [53] who scrutinized that CPEC would significantly improve opportunities in the areas. But CPEC has no influence on quality of life ($\beta$ = 0.029, $p > 0.05$) and also an insignificant influence on self-enhancement ($\beta$ = 0.004, $p > 0.05$) which did not support H2 and H3, respectively. These findings are different than [4] and [3] because they claimed CPEC as a game-changer and driver of quality life. However, considering the suggestion of [68], who claimed that CPEC would first develop the areas

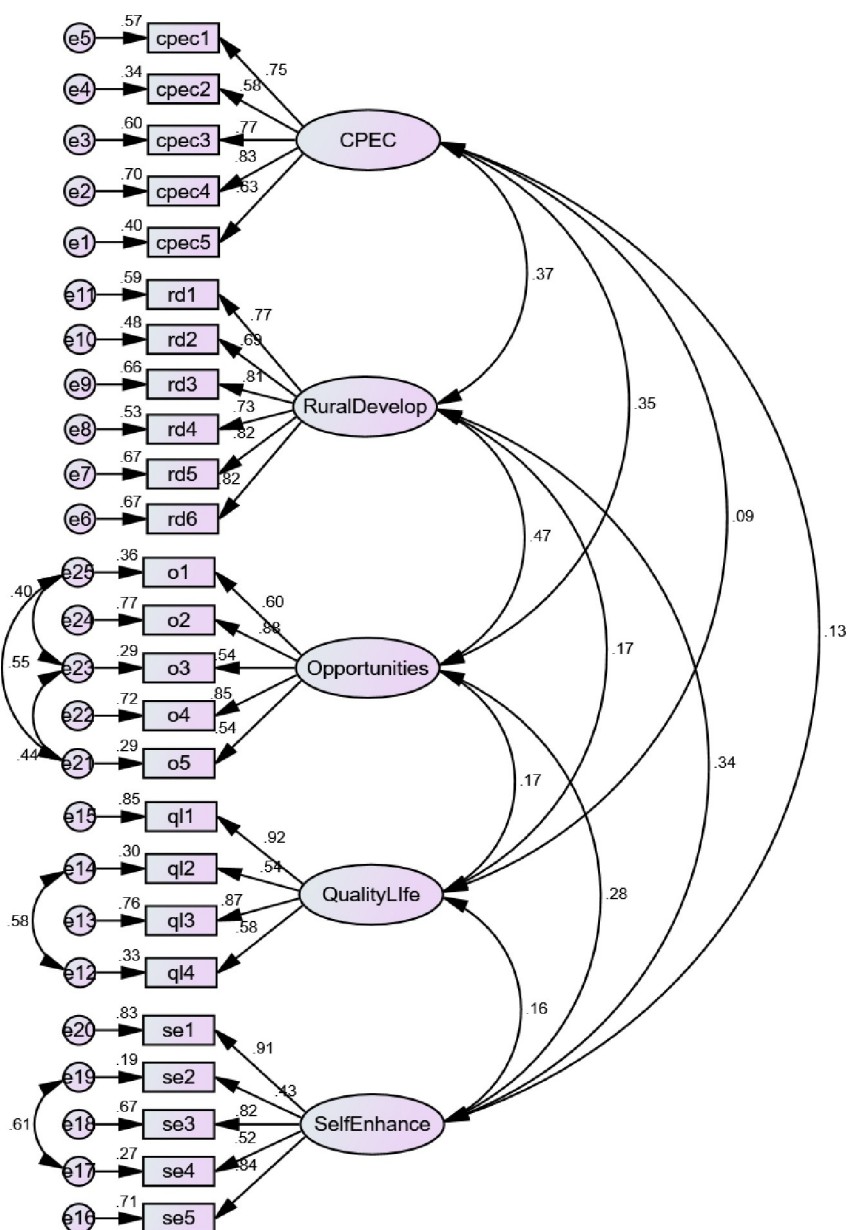

**Fig 6. Measurement model.**

which in turn would build an effective community, our findings also describe that CPEC would influence quality life and enhancement indirectly through the building and advancing the infrastructure. Building the CPEC has significant influence rural development (β = 0.369, p < 0.05) which supported H4. Our findings are in line with [14] who claimed that CPEC would improve infrastructure and would build the deprived areas. Rural development has a significant influence on opportunities (β = 0.399, p < 0.05), quality of life (β = 0.166, p < 0.05) and self-enhancement (β = 0.346, p < 0.05) which supported H5, H6 and H7, respectively. The indirect influence of CPEC on opportunities is significant (β = 0.147, p < 0.05) and the direct influence also remained significant which partially supported H8. The indirect effect of CPEC on quality of life is significant (β = 0.061, p < 0.05) and the direct influence become

**Table 4. Standardized regression weights.**

| Items | | Constructs | Estimate | AVE | √AVE | C.R. | Alpha |
|---|---|---|---|---|---|---|---|
| cpec5 | <--- | CPEC | 0.63 | 0.52 | 0.72 | 0.84 | 0.83 |
| cpec4 | <--- | CPEC | 0.83 | | | | |
| cpec3 | <--- | CPEC | 0.77 | | | | |
| cpec2 | <--- | CPEC | 0.58 | | | | |
| cpec1 | <--- | CPEC | 0.75 | | | | |
| rd6 | <--- | RuralDevelop | 0.82 | 0.60 | 0.77 | 0.90 | 0.90 |
| rd5 | <--- | RuralDevelop | 0.82 | | | | |
| rd4 | <--- | RuralDevelop | 0.73 | | | | |
| rd3 | <--- | RuralDevelop | 0.81 | | | | |
| rd2 | <--- | RuralDevelop | 0.69 | | | | |
| rd1 | <--- | RuralDevelop | 0.77 | | | | |
| ql4 | <--- | QualityLIfe | 0.58 | 0.56 | 0.75 | 0.83 | 0.50 |
| ql3 | <--- | QualityLIfe | 0.87 | | | | |
| ql2 | <--- | QualityLIfe | 0.54 | | | | |
| ql1 | <--- | QualityLIfe | 0.92 | | | | |
| se5 | <--- | SelfEnhance | 0.84 | 0.53 | 0.73 | 0.84 | 0.85 |
| se4 | <--- | SelfEnhance | 0.52 | | | | |
| se3 | <--- | SelfEnhance | 0.82 | | | | |
| se2 | <--- | SelfEnhance | 0.43 | | | | |
| se1 | <--- | SelfEnhance | 0.91 | | | | |
| o5 | <--- | Opportunities | 0.54 | 0.49 | 0.70 | 0.82 | 0.86 |
| o4 | <--- | Opportunities | 0.85 | | | | |
| o3 | <--- | Opportunities | 0.54 | | | | |
| o2 | <--- | Opportunities | 0.88 | | | | |
| o1 | <--- | Opportunities | 0.60 | | | | |

Note: AVE = Average Variance Extracted, CR = Composite Reliability, Alpha = Crobach Alpha.

insignificant which confirms the full mediating role of rural development and thus fully supported H9. Correspondingly, the indirect influence of CPEC on self-enhancement is significant ($\beta = 0.128$, $p < 0.05$) and the direct impact becomes insignificant which also fully supported H10. These results favor Ahmed [80], who claimed that CPEC would advance infrastructure in the deprived areas which would create favorable opportunities for the poor people and would improve their living standard.

R-square displays that 26%, 3% and 12% variation in opportunities, quality of life and self-enhancement are explained by CPEC in the presence of rural development as a mediator. Opportunities have satisfactory R-square values while the quality of life and self-enhancement indicate a low R-square that can be caused by excluding control variables such as age, educational background and households' size. Table 6 describes the summary of the hypothesized results.

## 6.3 Interview results

To avoid social desirability bias and to unleash the perceptions of uneducated women, we interviewed 32 rural women who have no educational qualifications but have an understanding of the CPEC. We asked the following questions to get comprehensive information about the perceptions of the rural women in the areas where CPEC projects were announced/started.

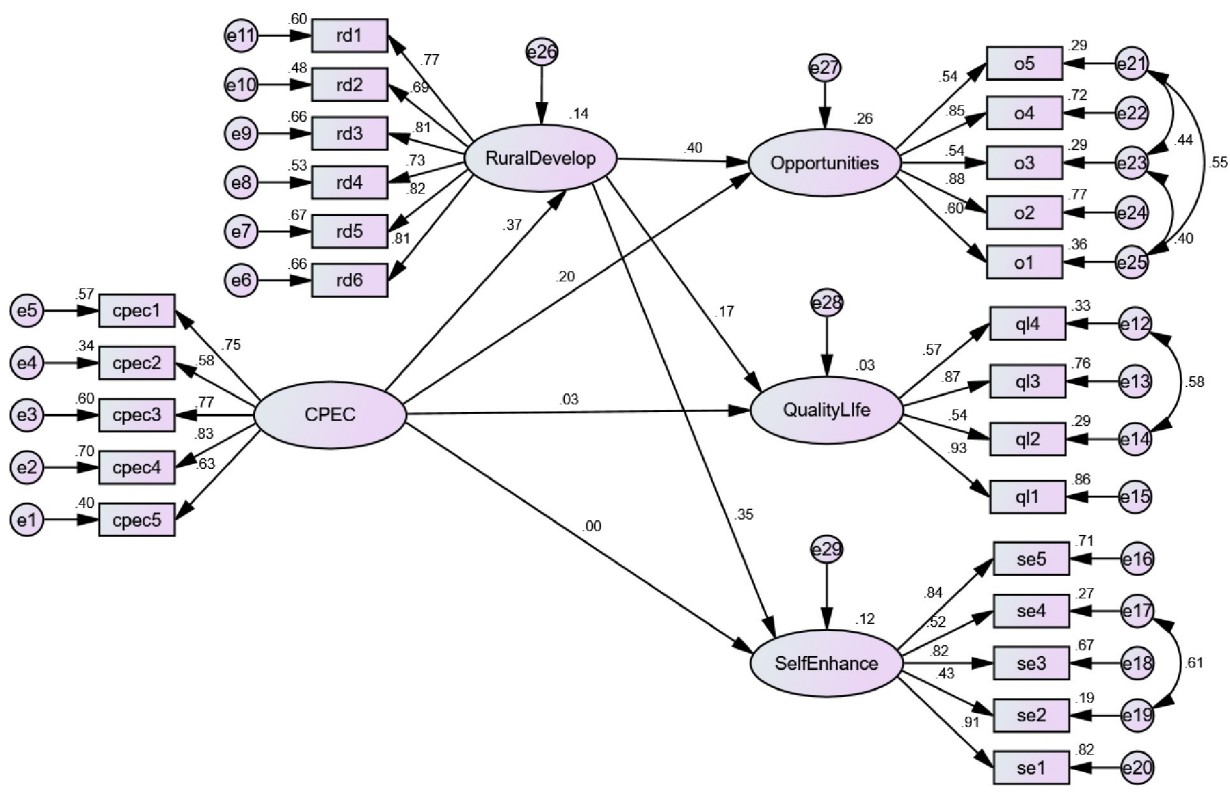

**Fig 7. Structural model estimation.**

1. What do you think of the CPEC?
   Ans: Out of a total 32 respondents, 24 said that the CPEC was a business and trade route to improve import and export between China and Pakistan. 5 respondents said that that it was a road for easy traveling of passengers between the countries. The rest 3 respondents said that it was a symbol of friendship between China and Pakistan.

2. Do you think the CPEC would create opportunities for you and what kinds and how?
   Ans: 19 respondents said "yes" and believed that CPEC would create several opportunities in the form of business development, environmental safety and tourism. However, they claimed that these opportunities would take place when if the government and public

**Table 5. Testing hypotheses.**

| Hypotheses | Direct effect | Indirect effect | Total effect |
|---|---|---|---|
| CPEC→ Opportunities | 0.201** | 0.147** | 0.348** |
| CPEC→ Quality of life | 0.029 | 0.061** | 0.091 |
| CPEC→ Self-Enhancement | 0.004 | 0.128* | 0.132* |
| CPEC→ Rural Development | 0.369** | - | 0.369** |
| Rural Development→ Opportunities | 0.399** | - | 0.399** |
| Rural Development→ Qualify of life | 0.166** | - | 0.166** |
| Rural Development→ Self-Enhancement | 0.346** | - | 0.346** |

Note = ** p value less than 0.005

* p value less than 0.05.

**Table 6. Hypothesized results.**

| Sr. No | Hypotheses | Remarks |
|---|---|---|
| H1 | *CPEC has a significant influence on perceived opportunities creation among rural women.* | S |
| H2 | *CPEC has a significant influence on the perceived quality of life among rural women.* | NS |
| H3 | *CPEC has a significant influence on perceived self-enhancement among rural women.* | NS |
| H4 | *The development of CPEC has a significant influence on rural development.* | S |
| H5 | *Rural development has a significant influence on perceived opportunities creation among rural women.* | S |
| H6 | *Rural development has a significant influence on the perceived quality of life among rural women.* | S |
| H7 | *Rural development has a significant influence on perceived self-enhancement among rural women.* | S |
| H8 | *The path between the development of CPEC and perceived opportunities creation among rural women is mediated by rural development.* | PS |
| H9 | *The path between the development of CPEC and perceived quality of life among rural women is mediated by the rural development* | PS |
| H10 | *The path between the development of CPEC and perceived self-enhancement among rural women is mediated by the rural development* | S |

S = supported, NS = Not supported, PS = partially supported.

bodies considered the development of the local community in terms of industries, roads, schools and hospitals. They also desired of starting new business, new startups and new ventures while exploiting the opportunities. The rest 13 respondents also intended that the route would create opportunities but they would not get benefits from it because of lack of trust in political and government authorities. These respondents claimed "the government has never listened to our voices for any opportunity regarding business and non-business. NPOs are working for our betterment but when the government involves and intervenes, their services and facilities do not reach us. The government only focuses on those people who have good and strong relationships with them, rather than other needy and poor people".

3. Will CPEC improve your quality of life?
   Ans: Answering this question, 16 respondents said "yes" and perceived that the CPEC would enhance their quality of life in several ways. They perceived that the new industries would improve the transport system in their areas and would save their traveling time—thereby resulting in betterment in the communities. 8 people said that urban people would like to visit the rural areas as tourist places, in this way tourism would boost in the areas that are a positive factor for the quality of life in the area. Additionally, 8 respondents said that due to the mega projects of CPEC in the rural areas, the government would focus on building relationships with the poor and rural communities which would favorably influence the quality of life.

4. Will CPEC positively influence your self-enhancement?
   Ans: 17 respondents are agreed with the question that CPEC would enhance their self-enhancement in decision making and daily affairs. They perceived that they would get a favorable status in society by initiating new business and participation in events near to their home. However, 15 disagreed and stated the reasons; culture resistance, family rigidity and low status in the society. They further said that despite several mega projects and initiatives by the government, we have been neglected by our society and the government. Our voice is considered poor in decision making and often neglected in daily decision making.

5. Do you think CPEC would improve infrastructure in the rural areas?
   Ans: 18 respondents said "yes" and desired that the mega projects of CPEC would improve the infrastructure of the areas which would create several opportunities for poor people. However, 14 people said that the government focuses only on specific areas rather than building and providing basic facilities for all the people equally.

6. What do you suggest to the government in terms of CPEC?
   Ans: There were 12 respondents who said that the government needs to hire and assign investment tasks to honest local bodies in the area rather than assigning to them to whom we have no access or they unequally distribute. There were 8 women who said that the government needs to provide our financial resources and low rental shops and places for doing business in the markets and commercial areas because we cannot afford the high rent of business shops. 4 women said we need equal rights in society in terms of employment in the industrial sector. The government needs to give us employment in the CPEC projects, so we would able to survive our family. 7 women said that we cannot access clean drinking water, electricity and schools were too away from us. Therefore, we strongly request to provide us these facilities in the mega projects. There was one respondent who said that the project team leaders and the government were advised to take care of the agriculture land and environmental safety. They also need to save the trees and farming land that were used for wheat and grains by our rural community.

## 6.4 Robustness test

To check the robustness of results, we used PROCESS macros in SPSS as recommended by Hayes and Preacher [81] and tested the mediating role of rural development between CPEC development and three dependent variables such as opportunities, quality of life and self-enhancement. The results reported in Table 7 show that the direct path ($\beta = 0.18$, $p < 0.01$) and indirect path ($\beta = 0.15$, $p < 0.01$, LLCI = 0.101, ULCI = 0.216) of CPEC development on opportunities are significant at 95% confidence interval. Furthermore, the results of the normal distribution theory test ($z = 5.60$, $P < 0.01$) approved that the indirect effect was significant which confirms that rural development partially mediates the relationship between CPEC development and opportunities, hence, H5 was partially supported. Anwar, Rehman [73] applied the same approach to test the robustness of the results and similar values were also reported in their study. The result of Table 8 show that the indirect effect of CPEC development on the quality of life (via rural development) was significant ($\beta = 0.08$, $p < 0.01$, LLCI = 0.027, ULCI = 0.142), while the direct effect of CPEC development on the quality of life is insignificant ($\beta = 0.03$, $p > 0.01$) at 95% confidence interval. Moreover the results of normal distribution theory test ($z = 2.62$, $P < 0.01$), agreed that the indirect effect was significant which suggests that *development* fully mediates the link between CPEC development and quality of life, thus H6 was fully accepted. Similarly, the direct path of CPEC development on self-enhancement was insignificant ($\beta = -0.02$, $p > 0.01$), whereas the indirect path of CPEC development on self-enhancement via rural development was significant ($\beta = 0.25$, $p < 0.01$, LLCI = 0.171, ULCI = 0.375). Moreover, the results of the normal distribution theory test ($z = 4.91$, $P < 0.01$) approved that the indirect path of CEPEC development on self-enhancement was significant so it indicates that rural development fully mediates the relationship between CPEC development and self-enhancement. Hence H7 is fully supported.

The R-square value (see Table 7) showed that CPEC development explained 30 percent variation in opportunities, 0.035 percent variation in the quality of life and 13 percent variation in self-enhancement in the presence of rural development. These results validated the results of SEM.

**Table 7. Results for main effect and mediation using Sobel test and bootstrapping.**

| The direct and total effect | | | | | |
|---|---|---|---|---|---|
| **Dependent: Opportunities, Independent: CPEC Development, Mediator: Rural Development** | | | | | |
| *Total effect* | *β* | *SE* | *p* | *t* | *R²* |
| Opportunities regressed on CPEC Development DV on IV | 0.33 | 0.04 | 0.00 | 7.39 | 0.151 |
| Rural Development regressed on CPEC Development MDV on IV | 0.52 | 0.06 | 0.00 | 7.81 | 0.169 |
| Opportunities regressed on Rural Development (controlling for CPEC Development) DV on MDV (controlling for IV) | 0.27 | 0.03 | 0.00 | 8.10 | 0.301 |
| *Direct effect* | | | | | |
| Opportunities regressed on CPEC Development (controlling for Rural Development) DV on IV (controlling for MDV) | 0.18 | 0.04 | 0.00 | 4.09 | |
| **Dependent: Quality of life, Independent: CPEC Development, Mediator: Rural Development** | | | | | |
| *Total effect* | *β* | *SE* | *p* | *t* | *R²* |
| Quality of life regressed on CPEC Development DV on IV | 0.11 | 0.06 | 0.08 | 1.70 | 0.009 |
| Rural Development regressed on CPEC Development MDV on IV | 0.53 | 0.6 | 0.00 | 7.81 | 0.169 |
| Quality of life regressed on Rural Development (controlling for CPEC Development) DV on MDV (controlling for IV) | 0.15 | 0.05 | 0.00 | 2.80 | 0.035 |
| *Direct effect* | | | | | |
| Quality of life regressed on CPEC Development (controlling for Rural Development) DV on IV (controlling for MDV) | 0.03 | 0.07 | 0.67 | 0.42 | |
| **Dependent: Self-Enhancement, Independent: CPEC Development, Mediator: Rural Development** | | | | | |
| *Total effect* | *β* | *SE* | *p* | *t* | *R²* |
| Self-Enhancement regressed on CPEC Development DV on IV | 0.23 | 0.09 | 0.01 | 2.50 | 0.020 |
| Rural Development regressed on CPEC Development MDV on IV | 0.53 | 0.06 | 0.00 | 7.81 | 0.169 |
| Self-Enhancement regressed on Rural Development (controlling for CPEC Development) DV on MDV (controlling for IV) | 0.47 | 0.07 | 0.01 | 6.37 | 0.137 |
| *Direct effect* | | | | | |
| Self-Enhancement regressed on CPEC Development (controlling for Rural Development) DV on IV (controlling for MDV) | -0.02 | 0.09 | 0.84 | -0.19 | |
| *indirect effect by normal theory distribution test* | | | | | |
| *Sobel test* | *β* | *SE* | *P* | *z* | |
| CPEC Development → Rural Development → Opportunities | 0.15 | 0.02 | 0.00 | 5.60 | |
| CPEC Development → Rural Development → Quality of life | 0.08 | 0.03 | 0.00 | 2.62 | |
| CPEC Development → Rural Development → Self-Enhancement | 0.25 | 0.05 | 0.00 | 4.91 | |
| *indirect effect results by method of Bootstrapping* | | | | | |
| Bootstrapping | Indirect effects | 95% -LLCI | | 95% -ULCI | |
| CPEC Development → Rural Development → Opportunities | 0.15 | 0.101 | | 0.216 | |
| CPEC Development → Rural Development → Quality of life | 0.08 | 0.027 | | 0.142 | |
| CPEC Development → Rural Development → Self-Enhancement | 0.25 | 0.171 | | 0.375 | |

Note: N = 232, 2,000-bootstrapping sample size, LLCI = Lower Limit Confidence Interval confidence interval, ULCI = Upper Limit Confidence Interval.

# 7. Discussion and conclusion

This study examined the benefits and advantages of CPEC for women in the rural areas of Pakistan. Although there are several studies that have examined the role of CPEC in economic development, employment and livelihood of citizens [5, 7, 15]. But, studies to examine how the CEPC would benefit rural women of Pakistan in terms of new opportunities, quality of life and self-enhancement are missing. Particularly, none of the studies has tested the mediating role of rural development between CPEC development and benefits for rural women in Pakistan. Therefore, this study contributes to the existing community development literature in the

**Table 8. Partial and full mediation.**

| Hypotheses | Direct effect | Indirect effect | Total effect | Mediation type |
|---|---|---|---|---|
| CPEC → Rural Development → Opportunities | 0.18(s) | 0.15(s) | 0.33(s) | Partial Mediation |
| CPEC → Rural Development → Quality of life | 0.03(ns) | 0.03(s) | 0.11(ns) | Full Mediation |
| CPEC → Rural Development → Self-Enhancement | -0.02(ns) | 0.25(s) | 0.23(s) | Full Mediation |

Note: s = Significant, ns = Not significant.

context of CPEC. This study contributes to the community development theory [22, 23] and social capital theory [25]. There is no denial of the widespread testing of these theories, however, they are still under-researched in the context of Asian economies. Moreover, in the context of the CPEC, none of the existing studies examined its benefits in the context of the social capital theory and community development theory. Our research confirms that the claims of the both theories and reveals that community development and wellbeing of the rural people are possible through developmental projects and government programs.

This study further revealed that the development of CPEC would significantly create useful opportunities for rural women in Pakistan. Rural development plays a partial mediating role between CPEC development and opportunities, which suggests that the creation of opportunities for the individuals are the results of the general rural development, and the effects of the CPEC development and translated into these opportunities only through the rural development. Our study supports Saad, Xinping [15] who claimed that CPEC was expected to create job opportunities for common citizens. Additionally, our study favors Shoukat, Ahmad [82] who claimed that CPEC would improve the rural infrastructure by encouraging international investors which in turn would create business opportunities. Considering the projects that are inaugurated under the CPEC, it was clear that the main emphasis of the CPEC was on improving infrastructures such as electricity, hospitals, schools and roads. These projects are intended to achieve long term benefits such as creating employment opportunities, environmental safety and educational improvement [83]. Lack of infrastructure and facilities, particularly electricity load shedding is the major barrier of business growth in rural areas [44]. Hence, CPEC is expected to play a very positive role in developing the rural areas that would overcome these issues, by generating new opportunities for the people in these areas. Therefore, CPEC can become a game-changer for rural women to start a desirable business and enjoy the infrastructure [84]. In contrary to the findings of Kanwal et al. (2019), our study demonstrated that rural women do not significantly perceive the positive change in their life via CPEC. Similarly, our study does not support the findings of Bano, Khayyam [57] who demonstrated that CPEC was perceived as a life-changer in the rural and urban areas of Pakistan. However, our study confirms that CPEC would indirectly improve the quality of life by rural development. When the rural areas are developed in terms of infrastructure, industry, electricity and school facilities, people connected with these areas would substantially improve their lifestyle. For instance, Ahmed [80] claimed that people from rural areas of Pakistan see a positive change in their lives after the area has been advanced due to the mega projects of CPEC.

We found that women in rural areas do not perceive self-enhancement that can be occurred directly through CPEC. Our results are slightly different than Saad, Xinping [15] who claimed that people directly perceive a positive change and high social status through CPEC. This difference comes from two major reasons. First, our respondents are different because Saad, Xinping [15] used empirical evidence of males and women while we examined the relationships for only women respondents. Second, Saad, Xinping [15] tested the direct influence of

CPEC on life change while our study tested and revealed an indirect influence of CPEC on life change via rural development, which also contribute to the existing literature by unearthing the mechanism through which the CPEC would affect the lives of the individuals. Our study shows that CPEC would not directly improve social status and self-enhancement until it developed the necessary infrastructure. However, our study favors Blanchard [61] who argued that CPEC would develop many areas of Pakistan and would connect them with big cities. In this way, the rural people would get urban services that would result in positive change in their social status and enhancement. However, the real change in their lives is when they see the improved infrastructure at their own areas.

Our study confirmed that rural women perceive many opportunities, quality of life and high social status and enhancement through rural development, which conforms with the earlier study Ali, Mi [60] who argued that infrastructure development benefits rural areas and poor people in several ways such as the provision of business opportunities, health facilitates and education institutions for them. In addition, during the interviews, many respondents claimed that the government did not provide equal employment opportunities in most of the cases. A few previous studies (e.g., [33, 43], 2007; [17]) also concluded similarly, that the government and public bodies ignore equality in rural areas and often neglect women and deprive them of their rights. However, rural women are hopeful and perceive that the CPEC would not discriminate between men and women and the same policies but would probably perform favorably than previous projects. Hence, the results obtained through the interview were aligned with the previous studies ([17, 43][17] who claimed that domestic violence and inequality in rural areas of Pakistan hinder women from progress and enhancement in the society.

## 7.1 Policy implications

Based on the insights obtained through the interviews and results of the study, we provide some policy implications for the policymakers and generally recommends that megaprojects of CPEC would create numerous opportunities for rural women of Pakistan. Fig 8 illustrates useful implications for community, project leaders (those who are currently parts of the CPEC projects in the areas) and the government. For instance, it indicates that the community should support CPEC activities and care for the projects in the areas. Project leaders need to provide the maximum number of employment opportunities to the rural areas people and those who are in need. The government and responsible authorities need to reconfigure their policies to benefit the maximum number of people in poor areas as described in the figure. It is further explained below.

We need to prepare the rural women to grab these opportunities arising out of these megaprojects. The government is required to support rural women and encourage them in entrepreneurial activities. The advancement in the infrastructure such as building schools, hospitals, roads and parks, would improve the lives of rural women. The government is recommended to invest money in rural development and build new hospitals, schools and parks in the rural areas enabling rural women to easily get the necessary information and basic facilities of life. Similarly, our study indicated that CPEC would not directly influence the self-enhancement and social status of the rural women in the areas but it would first develop infrastructure that would result in high social status. Therefore, policymakers need to focus on urbanization in rural areas and rural districts to connect them with big cities. Rural development would not only create new opportunities for rural women but would also encourage tourists from different areas that would result in positive change in the social interaction of rural women. The policymakers are required to:

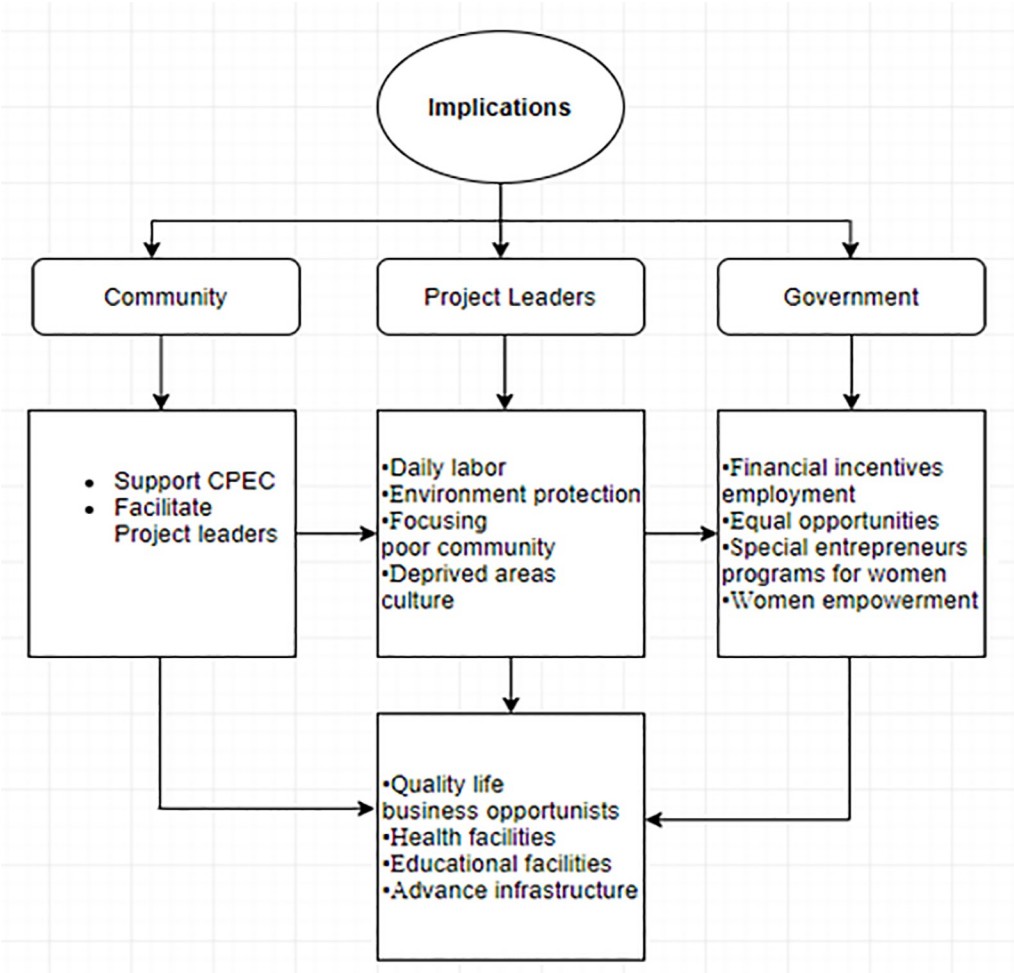

**Fig 8. Policy implications.**

- prioritize the rural development and build hospitals, schools and educational institutions to create new opportunities for rural women.

- improve connectivity between the rural areas and urban cities enabling women to access urban facilities;

- focus on rural development to modernize the poor villages and poor women enabling them to exploit new business opportunities.

- initiate special entrepreneurship programs such as training and certification in rural areas to boost the entrepreneurial activities of rural women.

- provide special incentives for rural development to reduce the social and agriculture challenges to improve the quality of life.

- improve the status of rural women in the society enabling them to confidently capitalize on the CPEC opportunities.

- empower the SMEDA to facilitate people in the business plan in the context of the rural regions and manage the initial resources, so they would able to easily start a new business.

- consider the establishment of some incubation centers for small businesses by providing the relevant necessary low-cost infrastructure.

- initiate microcredit and entrepreneurial finance programs for rural women, so they would easily launch their own business.

- create equal employment opportunities in CPEC for rural women, so all the rural women can get equal benefits; and

- Spur entrepreneurial activities in rural areas where more than 63% of people are living.

- Policymakers need to consider environmental issues in rural areas and are required to build policies for environmental protection to reduce air pollution in the areas.

## 7.2 Limitations and future studies

In terms of methodology, the first limitation of this research is that we collected data only from the women, and hence the results could not be generalized to the whole population. We used a structured questionnaire for data collection that was criticized for common method variance. Hence, there is advice for future researchers to conduct an in-depth interview with a few aged and educated women to gain useful information. Future research may collect evidence from rural and urban women and investigate their perceptions about the CPEC mega projects. Another important suggestion for future research is to survey Chinese women and explore their perception of CPEC to understand the benefits from a comparative perspective. We have not controlled the age, education and household dependency in this study, which can be controlled in future studies to reduce spuriousness results. We considered only a few social targets of these projects namely; opportunities, quality of life and self-enhancement in the present study. However, several other benefits and advantages of CPEC are expected for the economic and social welfare of households in China and Pakistan. For instance, Sher, Mazhar [8] claimed that CPEC would positively influence entrepreneurial activities and entrepreneurial intention among citizens. However, it is not yet explored either CPEC would influence the startup activities directly or indirectly via rural development, therefore, a further examination of the direct and indirect causal linkages between the CPEC and the startup activities would help understand the generating mechanism.

   We conclude that rural women significantly directly perceive new opportunities through CPEC, however, the opportunities are not directly affecting them, as they are effective indirectly through rural development. Particularly, the CPEC would not directly influence their quality of life and self-enhancement, but it would first develop infrastructure of the rural areas (e.g. housing, hospital, schools, roads and electricity etc.) which would result in the quality of life and self-enhancement. In other words, the rural development partially mediates the relationship between CPEC development and perceived opportunities while it fully mediates the relationships between CPEC development and quality of life as well as between CPEC development and self-enhancement. This research advises policymakers to accentuate on development of bucolic zones that would mend the living standards of poor communities in the neglected areas of Pakistan.

## Supporting information

**S1 Appendix.**
(DOCX)

## Acknowledgments

We are thankful to the rural areas people, especially women in participating the survey.

## Author Contributions

**Data curation:** Mariah Ijaz.

**Methodology:** Muhammad Usman Asghar.

**Project administration:** Muhammad Usman Asghar.

**Resources:** Liu Yamin.

**Software:** Liu Yamin.

**Writing – original draft:** Ahmad Saad.

**Writing – review & editing:** Ahmad Saad.

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
