## [Decision Letter · Decision Letter 0]

29 May 2020

PONE-D-20-12198

China-Pakistan Economic Corridor and its Impact on Rural Development and Human Life Sustainability; Observations from Rural Women

PLOS ONE

Dear Dr. Anwar,

Thank you for submitting your manuscript to PLOS ONE. After careful consideration, we feel that it has merit but does not fully meet PLOS ONE’s publication criteria as it currently stands. Therefore, we invite you to submit a revised version of the manuscript that addresses the points raised during the review process.

We look forward to receiving your revised manuscript.

Kind regards,

Bing Xue, Ph.D.

Academic Editor

PLOS ONE

Journal Requirements:

2. Please change "female” or "male" to "woman” or "man" as appropriate, when used as a noun.

3. Please provide additional details regarding participant consent. In the ethics statement in the Methods and online submission information, please ensure that you have specified (1) whether consent was informed and (2) what type you obtained (for instance, written or verbal, and if verbal, how it was documented and witnessed). If the need for consent was waived by the ethics committee, please include this information.

4. We note that [Figure(s) 4,7] in your submission contain [map/satellite] images which may be copyrighted. All PLOS content is published under the Creative Commons Attribution License (CC BY 4.0), which means that the manuscript, images, and Supporting Information files will be freely available online, and any third party is permitted to access, download, copy, distribute, and use these materials in any way, even commercially, with proper attribution. For these reasons, we cannot publish previously copyrighted maps or satellite images created using proprietary data, such as Google software (Google Maps, Street View, and Earth). For more information, see our copyright guidelines: http://journals.plos.org/plosone/s/licenses-and-copyright.

1.    You may seek permission from the original copyright holder of Figure(s) [4,7] to publish the content specifically under the CC BY 4.0 license. 

5. We suggest you thoroughly copyedit your manuscript for language usage, spelling, and grammar. If you do not know anyone who can help you do this, you may wish to consider employing a professional scientific editing service.  

6. Please amend your authorship list in your manuscript file to include author Muhammad Anwar

7. Please amend the manuscript submission data (via Edit Submission) to include author Ahmad Saad

8. PLOS requires an ORCID iD for the corresponding author in Editorial Manager on papers submitted after December 6th, 2016. Please ensure that you have an ORCID iD and that it is validated in Editorial Manager. To do this, go to ‘Update my Information’ (in the upper left-hand corner of the main menu), and click on the Fetch/Validate link next to the ORCID field. This will take you to the ORCID site and allow you to create a new iD or authenticate a pre-existing iD in Editorial Manager. Please see the following video for instructions on linking an ORCID iD to your Editorial Manager account: https://www.youtube.com/watch?v=_xcclfuvtxQ

9. Please upload a copy of Figure 14, to which you refer in your text on page 18. If the figure is no longer to be included as part of the submission please remove all reference to it within the text.

Reviewers' comments:

Reviewer's Responses to Questions

**Comments to the Author**

1. Is the manuscript technically sound, and do the data support the conclusions?

Reviewer #1: Partly

Reviewer #2: Partly

2. Has the statistical analysis been performed appropriately and rigorously? 

Reviewer #1: Yes

Reviewer #2: No

3. Have the authors made all data underlying the findings in their manuscript fully available?

Reviewer #1: No

Reviewer #2: Yes

4. Is the manuscript presented in an intelligible fashion and written in standard English?

Reviewer #1: No

Reviewer #2: No

5. Review Comments to the Author

Reviewer #1: The research topic and the overall idea is a very interesting one to say the least. CPEC and potential rural development is surely an important area when considering the successful achievement of SDGs. I have some questions and suggestions in regards to the submitted manuscript which are as follows:

Rural women have been the main focus in this research. The educational background provided by the authors suggests that there are 0.7% PhDs and 22.51% Masters/MS/M Phil while another 23.5% has Bachelor degree. According to the data presented in Table 1, about 45% of the women have a Bachelor degree or higher. This is an unusually highly educated population of women especially in a rural, under-developed settings. In Pakistan, such high level of education among women is a rarity even in urban areas and can only be seen in major cities. I believe that education level is self-reported by the participants and therefore, there is some level of bias in the provided responses. Since the authors are dealing with rural population, I would suggest explaining the participants that what is meant by these education levels and why accurate responses are vital to the research.

The measure of skewness & kurtosis is a good step and solidifies the descriptive statistics part.

In ‘Data Analysis and Results’, it is stated that “CPEC is significantly positively related with rural development (r = 0.411, p < 0.05), opportunities (r = 0.393, p < 0.05) and self-enhancement (= 0.143, p <0.05)”. Please re-consider this as significantly correlation requires at least an R-value of 0.5. For rural development and opportunities, it might be stated that moderate correlation has been found while for self-enhancement very weak correlation can be suggested. Same should be done for rural development and quality of life. Also, justification and support can also be provided to the current interpretation with references.

Similarly, when interpreting Structural Model results, kindly support the interpretation with references of one or more research publications.

In ‘Robustness Test’, it is stated that “the results of the normal distribution theory test (z = 5.60, P < 0.01) approved that the indirect effect is significant”. Please re-consider this as a z value of 5.60 is very high and means that there are outliers in the data which are causing such high z values. Also, justification and support can also be provided to the current interpretation with references.

I would suggest re-visiting the points highlighted here and then kindly review Table 8 for any possible changes to remarks section.

The ‘Policy Implications’ section presents some very significant points in relation to the research topic. I would suggest proposing a framework with the help of a flow chart or some other mechanism as deemed suitable. This is optional but can improve the quality of the manuscript significantly.

Overall, there are a number of spelling mistakes and additional proofreading is required.

I hope that these suggestions would assist the authors in improving the quality of the manuscript.

Reviewer #2: This paper studied the impacts of CPEC on rural development and human life sustainability. It is a useful and important study, which is well supported by the questionairs and well referenced. However, there are quite many questions about the study.

(1) The structure is quite confusing. For example, Section 4.2.1 comes after Section 3.3, and the Section 4.1 is after Section 4.2.13 ?

(2) It would be better that declare the theoretical framework part together, which may involve Section 2 and Section 4 in the manuscript.

(3) Section 6 is not really results, but description of data.

(4) I could not find the analysis part, but go strait from data description to conclusion.

(5) Why use so many semicolon punctuation in the manuscript? For example, the semicolon punctuation in the title.

(6) As you mentioned, there were 302 responses. However, in Section 6.5, most questions only get response from 19-32 respondents. Why?

6. PLOS authors have the option to publish the peer review history of their article (what does this mean?). If published, this will include your full peer review and any attached files.

Reviewer #1: No

Reviewer #2: No

---

## [Author Response · Author response to Decision Letter 0]

26 Jun 2020

Thank you very much for the Editor and Reviewers comments and their suggestions. After revising and incorporating the valuable suggestions and comments, we believe that the revised version of the manuscript is much approved and can be appropriate for publishing in the reputed journal, PLOS ONE. 

2. Please change "female” or "male" to "woman” or "man" as appropriate, when used as a noun.

Ans: We have incorporated the change through the manuscript and have used “women” instead of the word “female”. 

3. Please provide additional details regarding participant consent. In the ethics statement in the Methods and online submission information, please ensure that you have specified (1) whether consent was informed and (2) what type you obtained (for instance, written or verbal, and if verbal, how it was documented and witnessed). If the need for consent was waived by the ethics committee, please include this information.

Ans: Thank you for the comment. We have discussed how the survey is approved by the ethical committee for the data collection in the following way. 

We prepared a questionnaire and an in-depth interview for data collection that is approved by the research committee of National University of Science and Technology (NUST) for data collection. To get approval of the ethical committee, we presented them the written questionnaire with necessary information (cover letter, demographic information of individuals and questions of the main variables). After minor comments and discussion, they approved the survey for data collection.

4. We note that [Figure(s) 4,7] in your submission contain [map/satellite] images which may be copyrighted. All PLOS content is published under the Creative Commons Attribution License (CC BY 4.0), which means that the manuscript, images, and Supporting Information files will be freely available online, and any third party is permitted to access, download, copy, distribute, and use these materials in any way, even commercially, with proper attribution. For these reasons, we cannot publish previously copyrighted maps or satellite images created using proprietary data, such as Google software (Google Maps, Street View, and Earth). For more information, see our copyright guidelines: http://journals.plos.org/plosone/s/licenses-and-copyright.

1. You may seek permission from the original copyright holder of Figure(s) [4,7] to publish the content specifically under the CC BY 4.0 license. 

“I request permission for the open-access journal PLOS ONE to publish XXX under the Creative Commons Attribution License (CCAL) CC BY 4.0 (http://creativecommons.org/licenses/by/4.0/). Please be aware that this license allows unrestricted use and distribution, even commercially, by third parties. Please reply and provide explicit written permission to publish XXX under a CC BY license and complete the attached form.”Please upload the completed Content Permission Form or other proof of granted permissions as an "Other" file with your submission.

 Ans: Thank you for the comment. We have checked the figures again and realized that there is no need of the figures in the manuscript. Hence we have removed the figure from the paper. 

5. We suggest you thoroughly copyedit your manuscript for language usage, spelling, and grammar. If you do not know anyone who can help you do this, you may wish to consider employing a professional scientific editing service. 

Ans: We have proofread the manuscript from an experts in the relevant field and we believe that the manuscript is now free of grammatical mistakes. 

Ans: Dr. Wali Ur Rehman, Lecturer Essex Business School UK proofread the manuscript. His profile is here: https://scholar.google.com/citations?user=BzuJF0cAAAAJ&hl=en&oi=ao

6. Please amend your authorship list in your manuscript file to include author Muhammad Anwar

Ans: He is not an author in the manuscript. 

7. Please amend the manuscript submission data (via Edit Submission) to include author Ahmad Saad

Ans: The author is already added to the manuscript, and this is the final authors list. 

8. PLOS requires an ORCID iD for the corresponding author in Editorial Manager on papers submitted after December 6th, 2016. Please ensure that you have an ORCID iD and that it is validated in Editorial Manager. To do this, go to ‘Update my Information’ (in the upper left-hand corner of the main menu), and click on the Fetch/Validate link next to the ORCID field. This will take you to the ORCID site and allow you to create a new iD or authenticate a pre-existing iD in Editorial Manager. Please see the following video for instructions on linking an ORCID iD to your Editorial Manager account: https://www.youtube.com/watch?v=_xcclfuvtxQ

Ans: We have provided all the ORCID information of the author. 

9. Please upload a copy of Figure 14, to which you refer in your text on page 18. If the figure is no longer to be included as part of the submission please remove all reference to it within the text.

Ans: The figure is dropped as there was no specific information to be retained. 

Reviewers' comments:

Reviewer's Responses to Questions

Comments to the Author

1. Is the manuscript technically sound, and do the data support the conclusions?

Reviewer #1: Partly

Reviewer #2: Partly

2. Has the statistical analysis been performed appropriately and rigorously?

Reviewer #1: Yes

Reviewer #2: No

3. Have the authors made all data underlying the findings in their manuscript fully available?

Reviewer #1: No

Reviewer #2: Yes

4. Is the manuscript presented in an intelligible fashion and written in standard English?

Reviewer #1: No

Reviewer #2: No

5. Review Comments to the Author

Reviewer #1: The research topic and the overall idea is a very interesting one to say the least. CPEC and potential rural development is surely an important area when considering the successful achievement of SDGs. I have some questions and suggestions in regards to the submitted manuscript which are as follows:

Rural women have been the main focus in this research. The educational background provided by the authors suggests that there are 0.7% PhDs and 22.51% Masters/MS/M Phil while another 23.5% has Bachelor degree. According to the data presented in Table 1, about 45% of the women have a Bachelor degree or higher. This is an unusually highly educated population of women especially in a rural, under-developed settings. In Pakistan, such high level of education among women is a rarity even in urban areas and can only be seen in major cities. I believe that education level is self-reported by the participants and therefore, there is some level of bias in the provided responses. Since the authors are dealing with rural population, I would suggest explaining the participants that what is meant by these education levels and why accurate responses are vital to the research.

Ans: Thank you for the important comment. As mentioned in the manuscript that an un-educated person may not be able to properly predict potential benefits of a project, nor they are aware of governmental work. Therefore, we gathered evidence from educated women in the rural areas. In addition to this, we also conducted an in-depth interview with 32 those women who have never joined a school. It enabled us to reduce social desirability bias and facilitates us in getting reliable and true information for the research. 

The measure of skewness & kurtosis is a good step and solidifies the descriptive statistics part.

Ans: Thank you. 

In ‘Data Analysis and Results’, it is stated that “CPEC is significantly positively related with rural development (r = 0.411, p < 0.05), opportunities (r = 0.393, p < 0.05) and self-enhancement (= 0.143, p <0.05)”. Please re-consider this as significantly correlation requires at least an R-value of 0.5. For rural development and opportunities, it might be stated that moderate correlation has been found while for self-enhancement very weak correlation can be suggested. Same should be done for rural development and quality of life. Also, justification and support can also be provided to the current interpretation with references.

Ans: We have considering the suggestion and have discussed moderate and weak relationships where necessary. 

Similarly, when interpreting Structural Model results, kindly support the interpretation with references of one or more research publications.

Ans: We have supported the results with references. 

In ‘Robustness Test’, it is stated that “the results of the normal distribution theory test (z = 5.60, P < 0.01) approved that the indirect effect is significant”. Please re-consider this as a z value of 5.60 is very high and means that there are outliers in the data which are causing such high z values. Also, justification and support can also be provided to the current interpretation with references.

Ans: Thank you for pointing out the values to be checked. We have given relevant references who have mentioned similar values in the analysis. We have the references are supporting the values. However, if the ambiguity is still remained, let us know to revise the analysis. 

I would suggest re-visiting the points highlighted here and then kindly review Table 8 for any possible changes to remarks section.

Ans: Thank you for the suggestion, we have slightly revised the table. 

The ‘Policy Implications’ section presents some very significant points in relation to the research topic. I would suggest proposing a framework with the help of a flow chart or some other mechanism as deemed suitable. This is optional but can improve the quality of the manuscript significantly.

Ans: Thank for the suggestion. We have provided a figure and have discussed the implications articulately. 

Overall, there are a number of spelling mistakes and additional proofreading is required.

Ans: We have proofread the manuscript again to remove the grammatical mistake. 

I hope that these suggestions would assist the authors in improving the quality of the manuscript.

Reviewer #2: This paper studied the impacts of CPEC on rural development and human life sustainability. It is a useful and important study, which is well supported by the questionnaire and well referenced. However, there are quite many questions about the study.

(1) The structure is quite confusing. For example, Section 4.2.1 comes after Section 3.3, and the Section 4.1 is after Section 4.2.13 ?

Ans: Thank you for pointing out the ambiguity in the manuscript. We have revised the format and have corrected the numbering. 

(2) It would be better that declare the theoretical framework part together, which may involve Section 2 and Section 4 in the manuscript.

Ans: Thank you. We have incorporated the suggestion accordingly. 

(3) Section 6 is not really results, but description of data.

Ans: Incorporated. 

(4) I could not find the analysis part, but go strait from data description to conclusion.

Ans: Thank you. We have separated the results from the description of the data.

(5) Why use so many semicolon punctuation in the manuscript? For example, the semicolon punctuation in the title.

Ans: Thank you for pointing out the mistakes. We have proofread the manuscript and have removed the grammatical mistakes and unnecessary punctuations from the text. 

(6) As you mentioned, there were 302 responses. However, in Section 6.5, most questions only get response from 19-32 respondents. Why?

Ans: We used a structured questionnaire and collected data from 302 respondents in the rural areas. However, in addition to the questionnaire, we also conducted an in-depth interview with 32 uneducated women to articulate the results in a better way. To summarize, we use 302 structured questionnaires while 32 in-depth interviews for the data collection.

---

## [Decision Letter · Decision Letter 1]

24 Jul 2020

PONE-D-20-12198R1

China-Pakistan Economic Corridor and its Impact on Rural Development and Human Life Sustainability; Observations from Rural Women

PLOS ONE

Dear Dr. Saad,

Thank you for submitting your manuscript to PLOS ONE. After careful consideration, we feel that it has merit but does not fully meet PLOS ONE’s publication criteria as it currently stands. Therefore, we invite you to submit a revised version of the manuscript that addresses the points raised during the review process.

We look forward to receiving your revised manuscript.

Kind regards,

Bing Xue, Ph.D.

Academic Editor

PLOS ONE

Reviewers' comments:

Reviewer's Responses to Questions

**Comments to the Author**

1. If the authors have adequately addressed your comments raised in a previous round of review and you feel that this manuscript is now acceptable for publication, you may indicate that here to bypass the “Comments to the Author” section, enter your conflict of interest statement in the “Confidential to Editor” section, and submit your "Accept" recommendation.

Reviewer #1: All comments have been addressed

Reviewer #2: All comments have been addressed

2. Is the manuscript technically sound, and do the data support the conclusions?

Reviewer #1: Yes

Reviewer #2: Partly

3. Has the statistical analysis been performed appropriately and rigorously? 

Reviewer #1: Yes

Reviewer #2: Yes

4. Have the authors made all data underlying the findings in their manuscript fully available?

Reviewer #1: Yes

Reviewer #2: No

5. Is the manuscript presented in an intelligible fashion and written in standard English?

Reviewer #1: Yes

Reviewer #2: No

6. Review Comments to the Author

Reviewer #1: All of the concerns have been addressed. Just a few suggestions to reduce ambiguity and to improve the quality of this manuscript.

The sampling method needs to be mentioned in the methodology section. For instance, only educated women have been selected and approached for questionnaire needs to be mentioned like it is mentioned that other 32 women who never went to a school were interviewed. This will improve the understanding of the reader that how the sampling was conducted.

Reviewer #2: The structure is still very confusing. For example, Section 4.2.1 comes after Section 3.3, and the Section 4.1 is after Section 4.2.13, Section 1 is after Section 5?

7. PLOS authors have the option to publish the peer review history of their article (what does this mean?). If published, this will include your full peer review and any attached files.

Reviewer #1: No

Reviewer #2: No

---

## [Author Response · Author response to Decision Letter 1]

28 Jul 2020

Thank you once again the Editor and Reviewers for their time and comments since submission to finalization. We have incorporated all the comments in the revised manuscript. We hope the latest version is ready for publication. 

Grammatical and typing Mistake: We have proofread the paper again and have detected the grammatical and typing mistakes in the manuscript. We believe that the revised version is free of errors. 

6. Review Comments to the Author

Reviewer #1: All of the concerns have been addressed. Just a few suggestions to reduce ambiguity and to improve the quality of this manuscript.

The sampling method needs to be mentioned in the methodology section. For instance, only educated women have been selected and approached for questionnaire needs to be mentioned like it is mentioned that other 32 women who never went to a school were interviewed. This will improve the understanding of the reader that how the sampling was conducted.

Ans: Thank you for the useful comment. CPEC is initiated recently, it is a new project and is not a general project like others. Therefore, is argue that in general, an illiterate person may not able to understand and predict the benefits of the CPEC. During the survey and interview, we also acknowledge that most of the illiterate women do not know the CPEC. So it was difficult to receive relevant information for our research. Hence, we only interview those illiterate women who have some sort of knowledge of CPEC. 

“For instance, educated people have more knowledge and understanding of CPEC and other mega projects and can predict future benefits. However, to articulate the results in a better way, we also gathered information from 32 illiterate women. We interviewed only those illiterate women in rural areas who were aware of the CPEC. Since it was difficult to know the exact number of educated women in the areas. Therefore, we used a convenient sampling approach to complete the data collection. The convenient sampling method was also appropriate due to the wide geographic distance in these areas. Hence, both educated (for questionnaire) and illiterate (for interview) women were approached conveniently. Moreover, the respondents were informed in starting of the survey that “the survey is a volunteer and your information can help us in completing the research”. 

Reviewer #2: The structure is still very confusing. For example, Section 4.2.1 comes after Section 3.3, and the Section 4.1 is after Section 4.2.13, Section 1 is after Section 5?

Ans: Thank you once again for pointing out the mistakes, perhaps happened due to typing mistakes. We have corrected the numbering in the revised manuscript. 

Thank you.

---

## [Decision Letter · Decision Letter 2]

1 Sep 2020

PONE-D-20-12198R2

China-Pakistan Economic Corridor and its Impact on Rural Development and Human Life Sustainability; Observations from Rural Women

PLOS ONE

Dear Dr. Saad,

Thank you for submitting your manuscript to PLOS ONE. After careful consideration, we feel that it has merit but does not fully meet PLOS ONE’s publication criteria as it currently stands. Therefore, we invite you to submit a revised version of the manuscript that addresses the points raised during the review process.

We look forward to receiving your revised manuscript.

Kind regards,

Bing Xue, Ph.D.

Academic Editor

PLOS ONE

Reviewers' comments:

Reviewer's Responses to Questions

**Comments to the Author**

1. If the authors have adequately addressed your comments raised in a previous round of review and you feel that this manuscript is now acceptable for publication, you may indicate that here to bypass the “Comments to the Author” section, enter your conflict of interest statement in the “Confidential to Editor” section, and submit your "Accept" recommendation.

Reviewer #2: (No Response)

2. Is the manuscript technically sound, and do the data support the conclusions?

Reviewer #2: (No Response)

3. Has the statistical analysis been performed appropriately and rigorously? 

Reviewer #2: (No Response)

4. Have the authors made all data underlying the findings in their manuscript fully available?

Reviewer #2: (No Response)

5. Is the manuscript presented in an intelligible fashion and written in standard English?

Reviewer #2: (No Response)

6. Review Comments to the Author

Reviewer #2: The structure is still quite confusing.

For example, there are two sentions include literature review, such as Section 2 and Section 4.

7. PLOS authors have the option to publish the peer review history of their article (what does this mean?). If published, this will include your full peer review and any attached files.

Reviewer #2: No

---

## [Author Response · Author response to Decision Letter 2]

7 Sep 2020

Thank you once again the Editor and Reviewers for their time and comments 

We have received only one comment on the format of the manuscript. We have carefully reviewed the format and have merged some misleading numbering and headings. We believe that the revised format is free of vague and overlapping. 

We hope that the Editor will approve the paper and will provide his feedback soon. We are waiting to hear the good news soon, as to submit the paper for an award in our university. 

Thank you once again for your time and consideration.

---

## [Editor Report · Decision Letter 3]

9 Sep 2020

China-Pakistan Economic Corridor and its Impact on Rural Development and Human Life Sustainability; Observations from Rural Women

PONE-D-20-12198R3

Dear Dr. Saad,

We’re pleased to inform you that your manuscript has been judged scientifically suitable for publication and will be formally accepted for publication once it meets all outstanding technical requirements.

Kind regards,

Bing Xue, Ph.D.

Academic Editor

PLOS ONE
---

## [Editor Report · Acceptance letter]

11 Sep 2020

PONE-D-20-12198R3 

China-Pakistan Economic Corridor and its Impact on Rural Development and Human Life Sustainability; Observations from Rural Women 

Dear Dr. Saad:

I'm pleased to inform you that your manuscript has been deemed suitable for publication in PLOS ONE. Congratulations! Your manuscript is now with our production department. 

Kind regards, 

on behalf of

Professor Bing Xue 

Academic Editor

PLOS ONE